# Distributional Reinforcement Learning for Large Language Models

## Abstract

Actor-critic reinforcement learning for large language models (LLMs) typically relies on a scalar value function, discarding crucial information about potential returns. We propose a distributional actor-critic framework that learns the full distribution of returns to guide exploration more effectively. We find that in deterministic reasoning tasks, the spread of this learned distribution directly measures the model's confidence in its own value estimates. Our method harnesses this signal through an optimistic exploration bonus derived from the distribution's upper-tail variance, guiding the policy toward promising yet uncertain reasoning paths. This uncertainty-guided exploration promotes the discovery of diverse correct solutions, leading to substantial gains in pass@$k$ across challenging benchmarks. This result demonstrates a significant enhancement of the model's exploration effectiveness over strong baselines, which is complemented by consistent, albeit more modest, improvements in single-answer correctness.

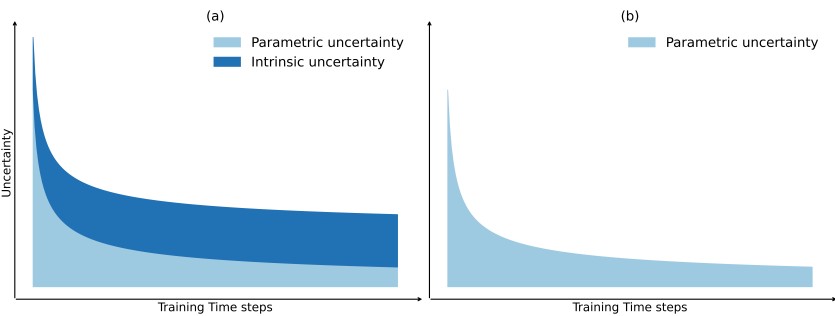

Figure 1: Uncertainty in general RL vs. LLM RL. (a) In classical RL, both parametric and intrinsic uncertainties coexist. (b) In LLM RL, the environment is deterministic and intrinsic uncertainty vanishes, leaving only parametric uncertainty as the driver of exploration.

## 1 Introduction

Large language models (LLMs) have recently achieved remarkable progress in dialogue, code generation, and mathematical reasoning. A key advance is their ability to generate long chains of thought (CoT), which enable step-by-step reasoning and have become a central benchmark for advanced AI (Wei et al., 2022; OpenAI, 2024; Guo et al., 2025). Reinforcement learning (RL) has been pivotal in this development, leveraging verifiable signals—such as mathematical correctness—to refine model behavior through structured exploration and credit assignment (Shao et al., 2024).

Recent RL applications in this area have explored several strategies. Methods like GRPO and DAPO operate without a value model, directly distributing trajectory-level rewards to tokens (Shao et al., 2024; Yu et al., 2025). In contrast, actor-critic approaches, notably PPO variants, learn a value function (critic) to provide finer-grained credit assignment at each step (Schulman et al., 2017; Yuan et al., 2025). While potentially more sample-efficient, the effectiveness of these actor-critic methods hinges on accurately estimating the value function over long, complex reasoning sequences (Yue et al., 2025b).

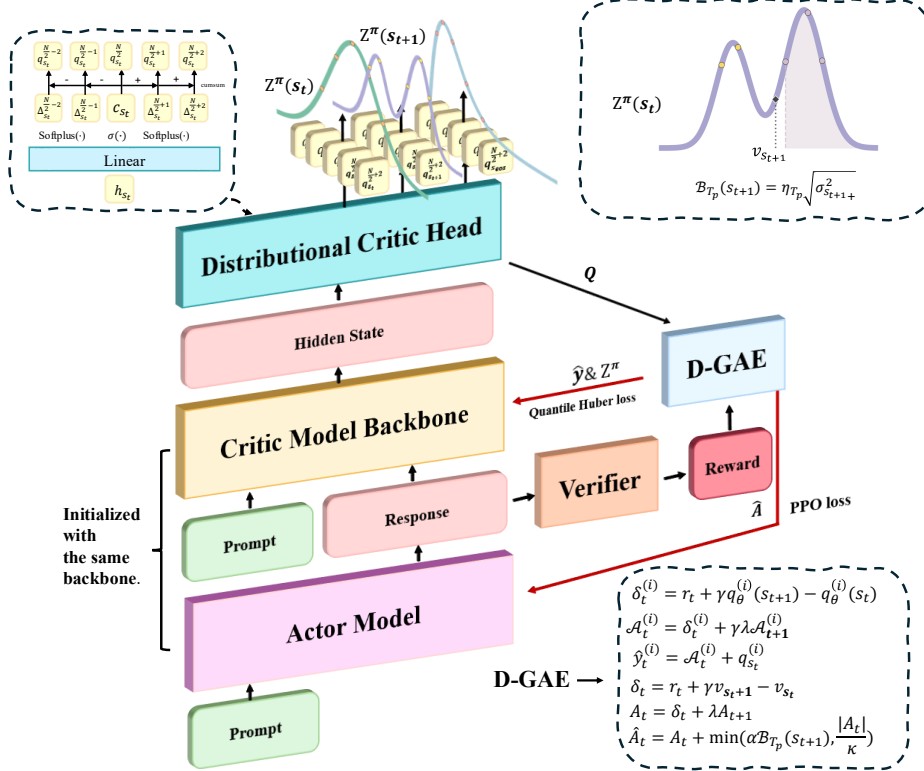

Figure 2: The actor maps a prompt to a response. The critic takes the prompt–response pair and, through a critic backbone (architecture identical to the actor's and initialized with the same parameters), produces hidden states $h_{s_t}$ that are fed into a distributional critic head to yield a return distribution $Z^\pi(s_t)$. Unlike PPO's scalar head generate a scalar $V_{s_t}$, our head outputs an ordered set of quantiles $\{q_\theta^{(n)}(s_t)\}$. These quantiles supervise critic learning via quantile Huber regression and drive actor updates through Distributional Generalized Advantage Estimation (D-GAE) with a Decaying Left–Truncated Variance (DLTV) exploration bonus. D-GAE returns (i) the multi-step target $\hat{y}_t$ for quantile regression and (ii) an optimistic advantage $\hat{A}_t$ for actor learning. Black arrows indicate inference; red arrows indicate training. Dashed boxes detail the center–delta quantile head (top left; for visual clarity the figure uses the shorthand $q_{s_t}^n$, which corresponds to $q_\theta^{(n)}(s_t)$ in the text), the truncated-variance bonus $B_{T_p}$ (top right; a plotting shorthand for $B_{T_{\text{step}}}$), and the D-GAE equations (bottom right).

In conventional actor–critic RL, the critic estimates a scalar value function, such as $Q^\pi(s, a)$ or $V^\pi(s)$, representing the expected cumulative return under policy $\pi$ (Sutton et al., 1998). While this expectation captures average performance, it ignores higher-order characteristics of the return distribution, including variance, skewness, and tail behavior. Such information can be critical for exploration and robust decision-making. Distributional reinforcement learning (DistRL) addresses this limitation by modeling the full distribution of returns, $Z^\pi(s, a)$, rather than only its expectation (Bellemare et al., 2017). Subsequent approaches, such as Quantile Regression DQN (QR-DQN) and Implicit Quantile Networks (IQN), provided practical methods for learning this distribution and demonstrated improved performance by capturing a richer signal for learning (Dabney et al., 2018b;a).

Crucially, the learned distribution of returns, $Z^\pi(s, a)$, implicitly captures two distinct sources of uncertainty (Mavrin et al., 2019). *Intrinsic uncertainty* stems from inherent randomness in the environment's transitions or rewards (Bellemare et al., 2017). *Parametric uncertainty* arises from the model's own limited knowledge and finite training data; it reflects the agent's ignorance about the true value function (Mannor et al., 2007). In most environments, like Atari games, the learned distribution conflates these two sources, making it difficult to isolate the model's estimation error (Mavrin

et al., 2019). However, *the setting of LLM-based reasoning presents a unique opportunity*. Here, the environment dynamics are effectively deterministic. A state $s_t$ (the previously generated tokens) combined with an action $a_t$ (the next token) deterministically produces the next state $s_{t+1}$. Rewards, such as correctness checks, are also typically deterministic. Consequently, intrinsic uncertainty vanishes. This implies that the entire spread of the learned value distribution can be interpreted as a pure measure of **parametric uncertainty**—a direct signal of the critic's own confidence in its estimates.

This insight allows us to repurpose the distributional critic from a general-purpose tool into a powerful quantifier of estimation uncertainty for guiding exploration. While existing PPO-based approaches for LLMs use a scalar value head that discards this information, we propose to leverage it. We introduce a **distributional actor–critic framework** that harnesses this parametric uncertainty as an exploration bonus. Specifically, we use the standard deviation of the upper tail of the distribution of $Z^\pi(s, a)$ as an intrinsic reward as that in (Mavrin et al., 2019), encouraging the agent to explore actions and states where it is uncertain about potentially high returns. This approach enables more directed and efficient exploration in complex, long-horizon reasoning tasks.

Our main contributions are summarized as follows:

- We reframe the role of distributional critics for LLMs by establishing that the deterministic nature of Chain-of-Thought reasoning eliminates intrinsic uncertainty. This key insight allows the learned return distribution to serve as a pure estimator of the model's parametric uncertainty.
- We operationalize this insight by introducing a parametric-uncertainty-guided exploration bonus, derived directly from the upper-tail variance of the learned value distribution. We integrate this bonus into a novel distributional actor-critic framework using a tailored Distributional Generalized Advantage Estimation (D-GAE) to effectively guide the policy. We show that this approach consistently improves pass@$k$ across reasoning benchmarks, outperforming strong actor–critic and value-model-free baselines. Qualitative analysis in section 5.3 further illustrates how our exploration mechanism promotes more structured reasoning.

## 2 PRELIMINARIES

A Markov Decision Process (MDP) is the standard mathematical framework for modeling reinforcement learning problems Sutton et al. (1998). It formally describes the interaction between an agent and its environment. An MDP is defined by the tuple $(\mathcal{S}, \mathcal{A}, P, R, \gamma)$, where $\mathcal{S}$ is the set of all possible states; $\mathcal{A}$ is the set of all possible actions; $P$ is the state transition probability function, with $P(s'|s, a)$ denoting the probability of transitioning from state $s$ to state $s'$ after taking action $a$; $R$ is the reward function, where $R(s, a)$ is the immediate reward; and $\gamma \in [0, 1)$ is the discount factor, which balances the importance of immediate and future rewards. Traditional reinforcement learning (RL) models the value of a state-action pair as a single scalar: the expected cumulative return. This expectation-based approach, however, collapses the entire distribution of potential outcomes into its mean, discarding valuable information about its variance and multimodality. Distributional reinforcement learning (DistRL) offers a more comprehensive framework by explicitly modeling the full probability distribution of the return (Bellemare et al., 2017; Dabney et al., 2018b; Qu et al., 2019).

Let $Z^\pi(s, a)$ denote the random variable for the discounted cumulative return initiated by taking action $a$ in state $s$ and subsequently following policy $\pi$. The objective in DistRL is to learn a model of the distribution of $Z^\pi(s, a)$, not merely its expectation $Q^\pi(s, a) = \mathbb{E}[Z^\pi(s, a)]$. The recursive nature of this distribution is captured by the **distributional Bellman equation**:

$$Z^\pi(s, a) \stackrel{D}{=} R(s, a) + \gamma Z^\pi(s', a'), \quad \text{where } s' \sim P(\cdot|s, a), \ a' \sim \pi(\cdot|s'),$$

where $\stackrel{D}{=}$ signifies equality in distribution. We can define a distributional Bellman operator $\mathcal{T}^\pi$ such that $Z^\pi(s, a)$ is a fixed point: $\mathcal{T}^\pi Z^\pi(s, a) \stackrel{D}{=} R(s, a) + \gamma Z^\pi(s', a')$.

A natural metric for comparing these return distributions is the Wasserstein distance (Villani, 2009). Consequently, a primary objective in DistRL is to minimize the distributional Bellman error, often measured by the 1-Wasserstein distance, $W_1(\mathcal{T}^\pi Z_\theta, Z_\theta)$, where $Z_\theta$ is the parameterized approximation to the return distribution induced by policy $\pi$.

However, directly minimizing the Wasserstein distance can be challenging. Quantile Regression DQN (QR-DQN) (Dabney et al., 2018b) provides an elegant solution by instead minimizing the **quantile regression loss** (Koenker & Bassett, 1978), which implicitly minimizes the Wasserstein distance between the empirical distributions.

In QR-DQN, the critic network is trained to predict a discrete set of $N$ quantiles, $\{q^{(i)}(s,a)\}_{i=1}^N$, corresponding to a predetermined set of probabilities $\{\tau_i\}_{i=1}^N$. These probabilities are typically chosen as a fixed set of uniformly spaced points in $[0, 1]$, e.g., $\tau_i = \frac{2i-1}{2N}$ for $i = 1, \ldots, N$.

For a given transition $(s, a, r, s')$, the set of target quantiles $\{y^{(j)}\}_{j=1}^N$ is defined as

$$\hat{y}^{(j)} = r + \gamma q^{(j)}(s', a').$$

The critic is trained by minimizing the quantile Huber loss over all pairwise differences between target and predicted quantiles:

$$\mathcal{L}_{\text{critic}} = \frac{1}{N^2} \sum_{i=1}^N \sum_{j=1}^N \rho_{\tau_i}^\kappa \big(\hat{y}^{(j)} - q_\theta^{(i)}(s,a)\big).$$

where $\rho_{\tau_i}^\kappa$ is the asymmetric quantile Huber loss:

$$\rho_{\tau_i}^\kappa \left( \hat{y}^{(j)} - q_\theta^{(i)}(s,a) \right) = \left| \tau_i - \mathbb{I}\left( \hat{y}^{(j)} - q_\theta^{(i)}(s,a) < 0 \right) \right| \mathcal{L}_\kappa \left( \hat{y}^{(j)} - q_\theta^{(i)}(s,a) \right),$$

$$\text{with} \quad \mathcal{L}_{\hat\kappa}(u) = \begin{cases} 0.5\, u^2, & |u| \le \hat\kappa, \\ \hat\kappa\,(|u| - 0.5\,\hat\kappa), & \text{otherwise.} \end{cases} \tag{1}$$

By training the critic to accurately predict the quantiles, QR-DQN captures a rich, multi-faceted representation of value, forming a foundational technique for modern DistRL algorithms. Notably, increasing the critic output from a single scalar to a moderate number of quantiles (e.g., $N = 51$) only enlarges the dimensionality of the output head, without requiring additional forward passes. In practice, this makes the extra computational cost modest relative to a standard scalar critic.

## 3 METHOD

We propose a distributional actor-critic framework designed to enhance the reasoning capabilities of language models. Our approach operates within an on-policy RL loop, akin to PPO, where trajectories are iteratively sampled and used for policy and value function updates. The core innovation lies in replacing the conventional scalar critic with a distributional critic that models the entire probability distribution of the state value return, denoted by the random variable $Z^\pi(s)$. This richer representation serves two synergistic purposes:

1. It provides stable, multi-step distributional targets for robust critic training, which we achieve using a novel **Distributional Generalized Advantage Estimation (D-GAE)** method.
2. It enables the quantification of model uncertainty, which we harness for exploration through a **Decaying Left-Truncated Variance (DLTV)** bonus.

A key distinction from the QR-DQN approach described in the Preliminaries is our focus on the state-value return $Z^\pi(s)$ rather than the state-action value return $Z^\pi(s, a)$. This adaptation is crucial for two reasons. First, in the context of Large Language Models, the action space (the entire vocabulary) is exceptionally large, making it computationally infeasible to model a separate distribution for every possible action. Second, in actor-critic architectures like PPO, the critic's primary role is to estimate the expected return $V^\pi(s)$ to compute an advantage function. **Therefore, we adapt the principles of distributional RL to model the distribution of the state-value return, the random variable $Z^\pi(s)$.** We use the notation $q^{(i)}(s)$ to represent the $i$-th quantile of this distribution.

Architecturally, we ensure that the predicted state-value quantiles $\{q^{(1)}(s) < q^{(2)}(s) < \cdots < q^{(N)}(s)\}$ are strictly ordered. Enforcing such non-crossing quantiles is important for obtaining a valid quantile function and is known to significantly improve stability in distributional RL (Zhou et al., 2020). The critic head first predicts a central quantile and then a sequence of non-negative deltas, which are cumulatively added to form the ordered quantiles. Further details are provided in Appendix C.

## 3.1 CRITIC UPDATE VIA DISTRIBUTIONAL GAE

To train the distributional critic with low-variance targets, we introduce Distributional Generalized Advantage Estimation (D-GAE). For each timestep $t$ in a trajectory, we first compute a set of $N$ quantile-wise TD errors based on the state value function:

$$\delta_t^{(i)} = r_t + \gamma q_\theta^{(i)}(s_{t+1}) - q_\theta^{(i)}(s_t). \tag{2}$$

Here, $q_\theta^{(i)}(s)$ denotes the $i$-th quantile of the value of state $s$, predicted by the critic network with parameters $\theta$. Analogous to standard GAE, we compute an exponentially-weighted sum of these future errors to form the distributional advantage for each quantile:

$$\mathcal{A}_t^{(i)} = \sum_{k=0}^{\infty} (\gamma\lambda)^k \delta_{t+k}^{(i)}, \quad \text{which can be computed recursively as } \mathcal{A}_t^{(i)} = \delta_t^{(i)} + \gamma\lambda \mathcal{A}_{t+1}^{(i)}. \tag{3}$$

The final multi-step learning target for each quantile, $\hat{y}_t^{(i)}$, is the sum of the current estimate and this advantage term:

$$\hat{y}_t^{(i)} = q_\theta^{(i)}(s_t) + \mathcal{A}_t^{(i)}. \tag{4}$$

With the predicted quantiles $\{q_\theta^{(i)}(s_t)\}$ and the constructed targets $\{\hat{y}_t^{(j)}\}$ (computed using the critic parameters $\theta^-$ from the rollout), we update the current critic parameters $\theta$ by minimizing the quantile Huber loss:

$$\mathcal{L}_{\text{critic}}(\theta) = \mathbb{E}_{\tau\sim\pi}\left[\frac{1}{T}\sum_{t=0}^{T-1}\frac{1}{N^2}\sum_{i=1}^{N}\sum_{j=1}^{N}\rho_{\tau_i}^{\kappa}\left(\hat{y}_t^{(j)} - q_\theta^{(i)}(s_t)\right)\right]. \tag{5}$$

where $\tau = (s_0, a_0, r_0, \ldots, s_T)$ denotes a full trajectory sampled from the current policy $\pi$.

## 3.2 UNCERTAINTY-GUIDED EXPLORATION WITH DLTV

A key advantage of our framework is the ability to extract a meaningful exploration bonus directly from the state-value critic. We design this bonus based on the principle of *optimism in the face of uncertainty*, a cornerstone of Upper Confidence Bound (UCB) algorithms (Auer et al., 2008; Lattimore & Szepesvári, 2020). This idea is closely related to prior distributional exploration approaches, including Distributional RL for efficient exploration (Mavrin et al., 2019; Zhou et al.), which leverages the spread of the value distribution to drive optimistic behavior . Our Decaying Left-Truncated Variance (DLTV) bonus, $\mathcal{B}_{T_{\text{step}}}(s)$, operationalizes this principle:

$$\mathcal{B}_{T_{\text{step}}}(s) = \eta_{T_{\text{step}}} \cdot \sigma_+(s). \tag{6}$$

It consists of an optimistic uncertainty estimate $\sigma_+(s)$ and a decaying schedule $\eta_{T_{\text{step}}}$.

The spread of the learned value distribution reflects the critic's parametric uncertainty. We quantify the uncertainty about a state's potential *upside* by computing the standard deviation over the upper half of its value quantiles:

$$\sigma_+(s) = \sqrt{\frac{1}{N/2}\sum_{i=N/2+1}^{N}\left(q_\theta^{(i)}(s) - q_\theta^{(N/2)}(s)\right)^2}. \tag{7}$$

This term measures the dispersion of optimistic outcomes relative to the median value, providing a clean signal for encouraging exploration into uncertain states. The decaying schedule is

$$\eta_{T_{\text{step}}} = c \cdot \sqrt{\frac{\log T_{\text{step}}}{T_{\text{step}}}}, \tag{8}$$

where $T_{\text{step}}$ is the training step and $c$ is a hyperparameter.

### 3.3 ACTOR UPDATE WITH OPTIMISTIC ADVANTAGE

Finally, we integrate the DLTV bonus into the actor's update. First, we compute a standard GAE advantage, $A_t(s_t, a_t)$.(for notational simplicity, we will write $A_t$ instead of $A_t(s_t, a_t)$ in what follows). This requires a scalar state-value estimate, $v_t$, which we obtain by averaging our predicted quantiles, a common convention in distributional RL (Barth-Maron et al., 2018):

$$v_t = \frac{1}{N} \sum_{i=1}^{N} q_\theta^{(i)}(s_t). \tag{9}$$

The standard advantage is then computed as

$$A_t = \delta_t + \gamma \lambda A_{t+1}, \quad \delta_t = r_t + \gamma v_{t+1} - v_t. \tag{10}$$

We then augment this advantage with our exploration bonus, which is a function of the *next state*, to create an **optimistic advantage estimate**, $\hat{A}_t$:

$$\hat{A}_t = A_t + \min\left( \alpha \cdot \mathcal{B}_{T_{\text{step}}}(s_{t+1}), \ \frac{|A_t|}{\kappa} \right). \tag{11}$$

The bonus $\mathcal{B}_{T_{\text{step}}}(s_{t+1})$ encourages exploration by favoring actions leading to uncertain states. The bonus is scaled by $\alpha$ and detached from the computational graph. It is also clipped relative to the advantage magnitude to maintain training stability. This final optimistic advantage $\hat{A}_t$ is then used in the PPO surrogate objective to update the actor's policy. This advantage shaping design is inspired by recent entropy-based advantage methods for exploratory reasoning in LLMs (Cheng et al., 2025), while our bonus is driven by distributional uncertainty rather than policy entropy. The full algorithm is given in Algorithm 1 (Appendix J).

## 4 RELATED WORK

**Reinforcement Learning for LLMs.** RL has become central to enhancing LLM reasoning, especially in long chain-of-thought (CoT) tasks (Wei et al., 2022; OpenAI, 2024; Guo et al., 2025; Shao et al., 2024). Value-model-free methods such as GRPO and DAPO avoid critics by redistributing trajectory-level rewards, offering stability and scalability for long sequences (Shao et al., 2024; Yu et al., 2025). In contrast, value-model-based methods like PPO provide finer credit assignment but often suffer collapse in long-CoT settings (Schulman et al., 2017; Yuan et al., 2025). Recent frameworks such as VAPO improve stability via value-regularization and length normalization (Yue et al., 2025b), highlighting renewed interest in critic-based training.

**Distributional Reinforcement Learning.** The advent of Distributional Reinforcement Learning (DistRL) marked a paradigm shift from modeling the expected return to capturing the full distribution of stochastic outcomes (Bellemare et al., 2017; Qu et al., 2019). Early works in this domain were predominantly value-based and designed for discrete action spaces. The seminal C51 algorithm (Bellemare et al., 2017) pioneered this approach by parameterizing the return distribution with a discrete set of uniformly spaced atoms over a predefined range. This idea was subsequently advanced by QR-DQN (Dabney et al., 2018b), which proposed using a discrete set of quantiles, and further generalized by IQN (Dabney et al., 2018a), which learns the full continuous quantile function, demonstrating enhanced performance and flexibility over fixed-atom representations. The extension of DistRL to policy-gradient methods suitable for continuous control tasks yielded algorithms such as Reactor (Gruslys et al., 2018) and Distributed Distributional Deterministic Policy Gradients (D4PG) (Barth-Maron et al., 2018). While D4PG demonstrated state-of-the-art performance, it inherits the representational limitations of C51, as its critic models the value distribution using a discrete categorical parameterization. This fundamentally constrains the expressive power of the critic, making it difficult to capture complex, non-uniform, or continuous return distributions. To address this, SDPG (Singh et al., 2022) introduced a sample-based distributional policy gradient method that models the return distribution via reparameterization, thereby avoiding the constraints of discrete representations. Our work is most closely related to (Mavrin et al., 2019) where the author used truncated variance bonuses to separate epistemic from intrinsic uncertainty. In our work, we develop more advanced D-GAE and leverage a key insight that transition and reward are deterministic in the LLM senario.

**Exploration in RL**. Efficient exploration is a central challenge in RL, with many strategies revolving around quantifying epistemic uncertainty. However, existing methods face significant hurdles when applied to large language models (LLMs). These challenges fall into two primary categories: prohibitive computational costs and fundamental theoretical limitations. In the first category, methods like Deep Ensembles (Osband et al., 2016; Lakshminarayanan et al., 2017), which train multiple independent models, are computationally infeasible for LLMs. Even more economical techniques such as Monte Carlo dropout Gal & Ghahramani (2016) impose a substantial overhead at inference time. In the second category, approaches like pseudo-counts (Bellemare et al., 2016; Ostrovski et al., 2017) are architecturally constrained, as their need for normalized probability densities precludes efficient batch processing. Others, like curiosity-driven exploration Burda et al. (2019); Pathak et al. (2017), lack a firm theoretical grounding for how the exploration bonus should be annealed over time. While the uncertainty Bellman equation (O'Donoghue et al., 2018) offers a principled framework, its practical application is undermined by the reliance on heuristics to estimate local uncertainty. The impracticality of these established techniques at the scale of LLMs highlights a critical gap, underscoring the need for a new paradigm in uncertainty-aware exploration.

## 5 EXPERIMENT

Our main experiments evaluate DistRL on challenging mathematical reasoning and SQL code generation tasks, showing that a distributional critic with uncertainty-guided exploration consistently improves both pass@$k$ and single-sample performance over strong RL baselines. In the appendix, we further (i) ablate the DLTV exploration bonusF.1, (ii) study the effect of the quantile number $N$ F.2, and (iii) conduct cross-domain evaluations to assess transfer beyond the RL training domainH.

### 5.1 EXPERIMENT SETUP

**Training Dataset and Backbone Models** We conduct experiments on the DAPO-Math-17k dataset released by the DAPO project (Yu et al., 2025). This dataset consists of 17K carefully curated mathematical reasoning problems collected from competition sources and web corpora. To assess the general applicability of our method, we use two distinct 7B-parameter backbone models from the Qwen2.5 series. Qwen2.5-Math-7B (Yang et al., 2024): A model specifically fine-tuned for mathematical reasoning. Qwen2.5-7B-Base (Qwen et al., 2025): The general-purpose dense base model. These selections allow for a comprehensive evaluation across both domain-specialized and generalist settings. For SQL generation, our experiments are conducted on Llama-3.1-8B-Instruct (Grattafiori et al., 2024). We trained on the Bird training set (Li et al., 2023).

**Baseline and Configuration** We compare against three strong baselines: GRPO (Shao et al., 2024), DAPO (Yu et al., 2025), and PPO (Schulman et al., 2017). All actor-critic experiments, including our method and the PPO baseline, are built upon the `veRL` framework (Sheng et al., 2024). To ensure a rigorous and fair comparison, our PPO baseline is augmented with several state-of-the-art techniques from DAPO and VAPO (Yue et al., 2025b), including Clip-Higher rewards, token-level loss calculation, critic pre-training, and group sampling. We additionally introduce two algorithms designed to encourage exploration as baselines: one uses entropy-based advantage shaping(Cheng et al., 2025), and the other incorporates intrinsic rewards via RND training (Gao et al., 2025). In practice, the number of quantiles $N$ is treated as a hyperparameter that controls the resolution of the value distribution approximation (as in QR-DQN). Based on preliminary ablation experiments that showed stable performance in this range, we set $N = 51$ and use this value for all experiments in the main text. Our method extends the PPO foundation with a distributional critic and a DLTV exploration bonus. Key hyperparameters follow standard settings, with full configurations provided in Appendix D.3.

**Evaluation Benchmarks and Metrics** To assess complex reasoning performance, we follow the evaluation protocol of Cheng et al. (2025). Specifically, we evaluate on AIME2024/2025 (MAA, 2025), MATH500 (Hendrycks et al., 2021), AMC (of America, 2023), OlympiadBench (He et al., 2024), College Math (Tang et al., 2024), and Minerva (Lewkowycz et al., 2022). We report two metrics under unbiased estimation: **pass@$k$**, which counts success if at least one of $k$ responses is correct, and **avg@$k$**, which measures the average correctness across the $k$ responses. Consistent with prior work, we follow standard decoding settings.(see I for details). BeyondAIME (Seed et al.,

Table 1: Performance on reasoning benchmarks with pass@$k$ (top) and avg@$k$ (bottom). Best results in **bold**, second best underlined, within each backbone group. $\Delta$ rows denote the gain for DistRL over PPO (green = gain, red = drop).

(a) pass@$k$ results

| | AIME25 pass@256 | AIME24 pass@256 | Minerva pass@16 | MATH500 pass@16 | OlympiadBench pass@16 | College pass@8 | Avg. |
|---|---|---|---|---|---|---|---|
| *Qwen2.5-Math-7B* | 53.3 | 70.0 | 50.4 | 88.6 | 56.7 | 44.2 | 60.5 |
| + GRPO | 50.0 | 76.7 | 64.0 | 91.8 | 59.7 | 49.2 | 65.2 |
| + DAPO | 56.7 | 76.7 | 66.9 | 92.0 | 60.9 | 48.3 | 66.9 |
| + DAPO w/ iMentor | 56.7 | 76.7 | 68.0 | 92.0 | 60.0 | 50.1 | 67.3 |
| + DAPO w/ Entropy Adv. | 60.0 | 83.3 | 66.5 | 91.4 | 57.6 | 48.5 | 67.9 |
| + PPO w/ Entropy Adv. | 60.0 | 76.7 | 68.0 | 91.2 | 62.0 | 50.4 | 68.1 |
| + PPO | 60.0 | 73.3 | 67.7 | 91.0 | 61.3 | 50.2 | 67.3 |
| + DistRL | 63.3 | 86.7 | 68.4 | 93.6 | 64.6 | 50.8 | 71.2 |
| $\Delta$ | +3.3 | +13.4 | +0.7 | +2.6 | +3.3 | +0.6 | +3.9 |
| *Qwen2.5-7B* | 53.3 | 56.7 | 50.0 | 84.2 | 52.7 | 47.2 | 57.3 |
| + PPO | 40.0 | 46.7 | 62.5 | 88.2 | 53.3 | 45.9 | 56.1 |
| + DistRL | 46.7 | 50.0 | 62.1 | 88.2 | 56.7 | 46.3 | 58.3 |
| $\Delta$ | +6.7 | +3.3 | -0.4 | 0.0 | +3.4 | +0.4 | +2.2 |

(b) avg@$k$ results

| | AIME25 avg@256 | AIME24 avg@256 | Minerva avg@16 | MATH500 avg@16 | OlympiadBench avg@16 | College avg@8 | Avg. |
|---|---|---|---|---|---|---|---|
| *Qwen2.5-Math-7B* | 4.4 | 10.7 | 16.9 | 47.5 | 20.4 | 22.1 | 20.3 |
| + GRPO | 11.2 | 28.7 | 41.8 | 79.0 | 40.3 | 42.0 | 40.5 |
| + DAPO | 16.5 | 31.9 | 41.0 | 81.5 | 41.4 | 41.0 | 42.2 |
| + DAPO w/ iMentor | 17.4 | 32.0 | 46.7 | 82.3 | 42.8 | 43.3 | 44.1 |
| + DAPO w/ Entropy Adv. | 17.2 | 33.3 | 44.5 | 80.9 | 41.4 | 41.6 | 43.2 |
| + PPO w/ Entropy Adv. | 17.1 | 31.9 | 45.0 | 81.0 | 42.9 | 43.0 | 43.4 |
| + PPO | 16.4 | 31.3 | 44.7 | 81.0 | 42.8 | 42.7 | 43.1 |
| + DistRL | 16.8 | 32.3 | 45.8 | 82.5 | 43.6 | 43.5 | 44.1 |
| $\Delta$ | +0.4 | +1.0 | +1.1 | +1.5 | +0.8 | +0.8 | +1.0 |
| *Qwen2.5-7B* | 2.2 | 5.2 | 11.4 | 43.2 | 15.6 | 24.5 | 17.0 |
| + PPO | 8.1 | 16.3 | 37.1 | 75.6 | 34.8 | 38.6 | 35.1 |
| + DistRL | 10.3 | 16.2 | 37.3 | 76.3 | 36.2 | 39.0 | 35.9 |
| $\Delta$ | +2.2 | -0.1 | +0.2 | +0.7 | +1.4 | +0.4 | +0.8 |

2025) is a newly released large-scale and highly challenging benchmark of 100 problems. We additionally report pass@$k$ to measure how effectively the model can discover correct solutions with multiple samples, serving as an indicator of exploration efficiency. For SQL generations, we evaluated performance on the Bird and Spider test sets (Li et al., 2023; Yu et al., 2019).

## 5.2 RESULTS

**Main Benchmarks.** Table 1 summarizes results on six reasoning benchmarks. Overall, our proposed *DistRL* method consistently delivers the best performance on the Qwen2.5-Math-7B backbone, improving upon strong baselines such as PPO, GRPO, and DAPO. In terms of pass@$k$, DistRL achieves an average score of 71.2, compared to 67.3 for PPO, while avg@$k$ rises from 43.5 to 44.1. These improvements are particularly pronounced on difficult datasets: for example, DistRL raises AIME25 pass@*256* from 60.0 to 63.3 and OlympiadBench pass@*16* from 61.3 to 64.6. The consistent gains across both pass@$k$ and avg@$k$ indicate that distributional critics provide more reliable exploration signals, enabling stronger reasoning under limited samples.

For completeness, we also report results on the general-purpose Qwen2.5-7B backbone, confirming that our method generalizes beyond math-specialized models.

**BeyondAIME Benchmark.** We further evaluate on BeyondAIME, a math benchmark without training contamination, and the pass@$k$ results in Figure 3 demonstrate enhanced reasoning ability.

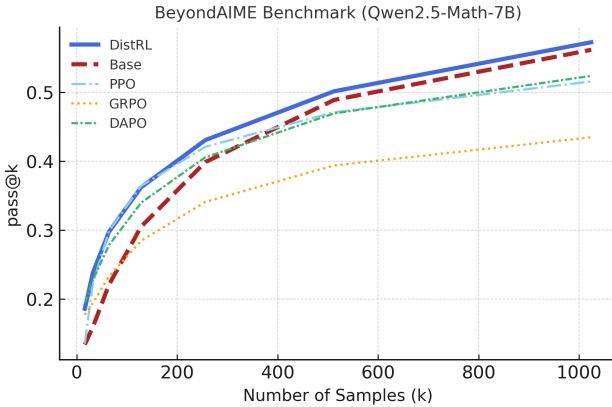

Figure 3: pass@$k$ results on BeyondAIME benchmark (Qwen2.5-Math-7B). DistRL consistently outperforms PPO and other baselines across different sample sizes, demonstrating higher exploration effectiveness.

Table 2: Performance on SQL generation benchmarks with greedy sampling and pass@$k$. Best results in **bold**, second best underlined, within each backbone group. $\Delta$ rows denote the gain for DistRL over PPO (green = gain, red = drop).

| Model | **Bird** (in domain) | | | **Spider** (out of domain) | | | Avg |
|---|---|---|---|---|---|---|---|
| | Greedy | *Pass@8* | *Pass@16* | Greedy | *Pass@8* | *Pass@16* | |
| *Llama-3.1-8B-Instruct* | 42.4 | 68.5 | 75.1 | 69.0 | **91.0** | **94.6** | 73.4 |
| + GRPO | 60.7 | 72.2 | 74.6 | 74.7 | 81.0 | 82.9 | 74.4 |
| + DAPO | 63.2 | 73.9 | 75.9 | 76.8 | 86.1 | 87.2 | 77.2 |
| + DAPO w/ iMentor | 62.7 | 73.9 | 76.1 | 77.2 | 86.4 | 88.2 | 77.4 |
| + DAPO w/ Entropy Adv. | 62.3 | 73.2 | 75.9 | 77.5 | 86.1 | 87.6 | 77.1 |
| + PPO w/ Entropy Adv. | 62.0 | 73.3 | 76.0 | 77.6 | 86.3 | 87.9 | 77.2 |
| + PPO | 61.4 | 73.1 | 74.9 | 77.0 | 85.2 | 86.6 | 76.3 |
| + DistRL (ours) | **63.5** | **75.0** | **77.4** | **81.2** | 87.5 | 88.9 | **78.9** |
| $\Delta$ | +2.1 | +1.9 | +2.5 | +4.2 | +2.3 | +2.3 | +2.6 |

Our DistRL-enhanced model consistently outperforms all baselines across most pass@$k$ settings on the Qwen2.5-Math-7B backbone. This superiority is particularly pronounced at high values of $k$, which approximate the model's exploration effectiveness. For instance, DistRL achieves a pass@1024 score of 58.0, significantly surpassing the strong PPO baseline's 52.0. This sustained advantage under extensive sampling indicates that our distributional exploration strategy does more than just improve sampling efficiency—it genuinely elevates the model's problem-solving potential.

These findings speak directly to the discussion raised by Yue et al. (2025a). Our results on this uncontaminated benchmark suggest that RL can help models more reliably express their latent reasoning capabilities. By improving the exploration process and enabling access to a broader set of viable reasoning trajectories, DistRL allows the underlying model to achieve stronger reasoning performance than what is typically observed under standard decoding.

**SQL Benchmark.** To assess generality beyond mathematical reasoning, we evaluate our method on SQL generation using BIRD (in-domain) and Spider (out-of-domain). As shown in Table 2, DistRL consistently outperforms PPO, GRPO, and DAPO across greedy decoding and pass@$k$. Notably, it yields +2.1–2.5 gains on BIRD and larger +2.3–4.2 gains on Spider, where cross-domain generalization is more challenging. These improvements indicate that uncertainty-aware distributional value modeling benefits structured, program-like reasoning tasks, demonstrating that our method extends beyond math to broader symbolic reasoning settings.

### 5.3 QUALITATIVE ANALYSIS OF THE EXPLORATION MECHANISM

To qualitatively understand how our uncertainty-guided exploration shapes the model's behavior, we visualize the sentences that receive the highest exploration bonuses at different stages of training.

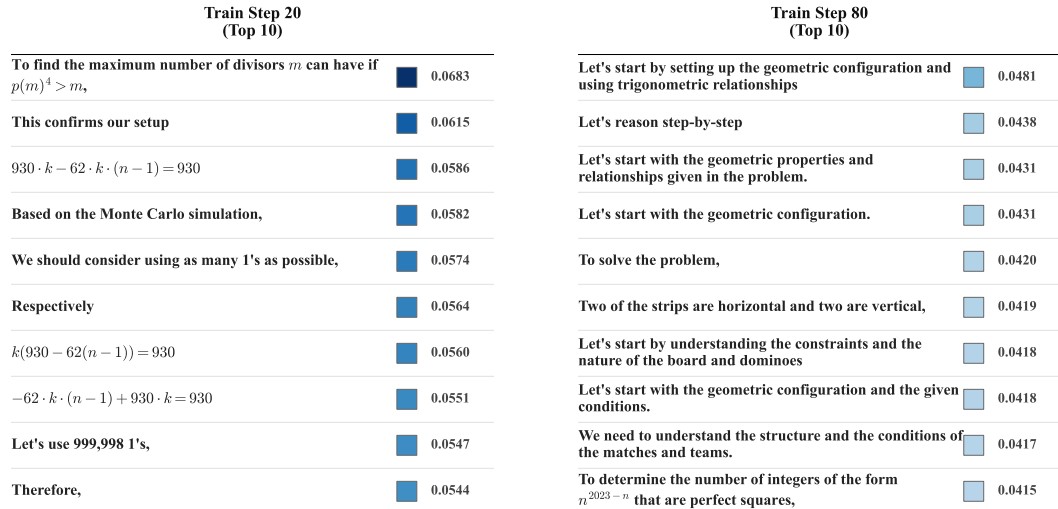

Figure 4: Top-10 high-variance sentences at different training steps. Early in training (step 20), the variance bonus highlights scattered problem setup and numeric fragments. Later (step 80), it shifts to structured reasoning instructions, indicating more stable and interpretable reward allocation.

Figure 4 contrasts the top-10 high-bonus sentences from an early checkpoint (step 20) with those from a more mature checkpoint (step 80).

At the early stage of training (step 20), the bonus predominantly targets sentences related to **foundational components** of the reasoning process. These include low-level problem setups and computational fragments, such as "*to find the maximum number of divisors ...*" and "*[930 cdot k - 62 ... ]*". This suggests that the model is initially uncertain about basic problem formulation and uses the exploration signal to solidify its understanding of diverse entry points into the solution space. A distinct shift occurs as training progresses. By the later stage (step 80), the high-bonus sentences are no longer computational but rather **strategic and structural** in nature. The focus moves to meta-reasoning prompts that organize the solution pathway, such as "*let's start by setting up the geometric configuration*" and "*let's reason step-by-step*". This transition indicates that the distributional critic has learned to incentivize the exploration of high-level reasoning strategies, rather than isolated calculations.

In summary, this visualization provides compelling evidence that our exploration mechanism is not random but **programmatic**. It progressively guides the model's learning process, shifting its focus from surface-level, trial-and-error computations toward the construction of coherent and structured reasoning strategies. This observed evolution from computational to strategic uncertainty aligns perfectly with the quantitative improvements seen in our benchmark results.

## 6 CONCLUSION

We presented a distributional actor–critic framework for LLM reasoning, featuring a center–delta quantile head, distributional GAE, and a DLTV-based exploration bonus. Experiments on diverse mathematical reasoning benchmarks show that our method consistently outperforms PPO, GRPO, and DAPO, achieving state-of-the-art results on Qwen2.5-Math-7B and further gains on the challenging BeyondAIME benchmark. These results highlight the effectiveness of distributional critics in enhancing both stability and exploration.

## 7 ETHICS STATEMENT

This work adheres to the ICLR Code of Ethics. In this study, no human subjects or animal experimentation was involved. All datasets used were sourced in compliance with relevant usage guidelines, ensuring no violation of privacy. We have taken care to avoid any biases or discriminatory outcomes in our research process. No personally identifiable information was used, and no

experiments were conducted that could raise privacy or security concerns. We are committed to maintaining transparency and integrity throughout the research process.

## 8 REPRODUCIBILITY STATEMENT

We have made every effort to ensure that the results presented in this paper are reproducible. All code and datasets have been made publicly available in an anonymous repository to facilitate replication and verification. The experimental setup, including training steps, model configurations, and hardware details, is described in detail in the paper. Our work is reproducible, and the code is openly available at: `https://anonymous.4open.science/r/verl_DistRL-D384/`.

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

## A  THE USE OF LARGE LANGUAGE MODELS (LLMS)

LLMs were employed during the writing of this paper to polish the text and correct grammatical errors. The prompt used was: "Please detect and correct any grammatical errors in the following text, and polish it to enhance its academic expression. <text>"

## B  ENTROPY REGULARIZATION AS A GENERALIZED $\epsilon$-GREEDY EXPLORATION

We provide a mathematical derivation showing that entropy regularization corresponds to a softmax exploration strategy. This can be interpreted as a generalized form of $\epsilon$-greedy exploration that intelligently allocates exploration probability based on the relative quality of suboptimal actions.

**Entropy-regularized policy improvement.**  Given a state $s$ and advantage estimates $A(s, a)$ for actions $a \in \mathcal{A}$, consider the entropy-regularized optimization:

$$\pi^\star = \arg\max_{\pi(\cdot|s)} \sum_a \pi(a|s) A(s, a) + \beta H(\pi(\cdot|s)),$$

where $H(\pi) = -\sum_a \pi(a|s) \log \pi(a|s)$ and $\beta > 0$ is the entropy coefficient. The well-known solution is the Boltzmann/softmax distribution:

$$\pi_\beta(a|s) = \frac{\exp(A(s, a)/\beta)}{\sum_{b \in \mathcal{A}} \exp(A(s, b)/\beta)}.$$

**Connection to $\epsilon$-greedy.**  Let $a^\star \in \arg\max_a A(s, a)$ and denote the advantage gaps $\Delta_a = A(s, a^\star) - A(s, a) \geq 0$. The probability of selecting the optimal action is

$$p_\star = \pi_\beta(a^\star|s) = \frac{1}{1 + \sum_{a \neq a^\star} \exp(-\Delta_a/\beta)}.$$

We can define a state- and value-dependent exploration probability $\epsilon_\beta(s) = 1 - p_\star$. This allows us to decompose the policy as:

$$\pi_\beta(\cdot|s) = (1 - \epsilon_\beta(s)) \delta_{a^\star} + \epsilon_\beta(s) q_\beta(\cdot|s),$$

where $q_\beta(a|s) \propto \exp(-\Delta_a/\beta)$ is a probability distribution over the set of suboptimal actions.

This formulation reveals that softmax exploration is a generalized form of $\epsilon$-greedy. However, unlike the standard $\epsilon$-greedy rule, its exploration is not uniform. The distribution $q_\beta$ intelligently assigns higher probability to suboptimal actions that are closer to optimal (i.e., having a smaller advantage gap $\Delta_a$). Only under the strong and often unrealistic condition that all suboptimal actions are equally bad ($\Delta_a \approx$ const. for $a \neq a^\star$) does $q_\beta$ approach a uniform distribution, making the strategy resemble standard $\epsilon$-greedy. Thus, entropy regularization typically leads to a more efficient exploration strategy than its uniform counterpart.

### B.1 ORTHOGONALITY AND COMPLEMENTARITY OF DLTV AND ENTROPY-BASED EXPLORATION

Our proposed exploration bonus, DLTV (Decaying Left-Truncated Variance), operates on a principle that is orthogonal yet complementary to entropy regularization. They address different facets of the exploration problem.

**Entropy regularization: policy-centric exploration.** Entropy-based exploration enforces *distributional smoothness* on the policy $\pi(\cdot|s)$. It provides an *undirected* exploration drive by preventing the policy from collapsing prematurely to a deterministic choice. This ensures a baseline level of global coverage, but it is agnostic to the agent's underlying uncertainty about the action values.

**DLTV exploration: value-centric exploration.** In contrast, DLTV provides a *directed* exploration bonus by modifying the *advantage estimates* directly. This encourages the policy model to specifically probe actions whose long-term values are poorly estimated.

**Orthogonal but complementary.** When combined, the two mechanisms contribute additively to the policy update objective. The effective advantage shaping for the policy gradient update can be expressed as:

$$A_{\text{total}}(s, a) = A(s, a) + \underbrace{B_{\text{DLTV}}(s, a)}_{\text{Value Uncertainty Bonus}} + \underbrace{\beta(-\log \pi_\theta(a|s))}_{\text{Policy Stochasticity Bonus}} .$$

These terms are orthogonal in the sense that they originate from distinct information sources:

- The **DLTV bonus** is a value-centric signal derived from the *critic's uncertainty*, directing exploration towards actions where knowledge is lacking.
- The **entropy bonus** is a policy-centric regularizer derived from the *policy's own stochasticity*, maintaining global diversity.

At the same time, they are complementary because their combination yields a more balanced exploration strategy: DLTV provides efficiency through uncertainty-guided exploration, while entropy regularization ensures sufficient coverage across the action space.

## C DISTRIBUTIONAL VALUE HEAD

To ensure that the predicted state-value quantiles $\{q_\theta^{(1)}(s) < q_\theta^{(2)}(s) < \cdots < q_\theta^{(N)}(s)\}$ form a valid and strictly ordered quantile function, we adopt a *center–delta cumulative* parameterization. The head produces (i) a central reference quantile and (ii) a sequence of non-negative deltas whose cumulative sums define the remaining quantiles. This guarantees monotonicity *by construction*, without requiring any sorting or auxiliary constraints during training.

**Architecture.** Given the critic backbone hidden state $h_s \in \mathbb{R}^d$ for state $s$, the value head consists of two linear projections:

$$c_s = W_c h_s \in \mathbb{R}, \tag{12}$$

$$\Delta_s = W_\Delta h_s \in \mathbb{R}^{N-1}, \tag{13}$$

where $N$ is the number of quantiles and $m = \lfloor N/2 \rfloor$ denotes the index of the center quantile. Both projections are implemented as bias-free linear layers with dropout for regularization.

**Non-negative deltas and strict ordering.** Following the design philosophy used in distributional modeling with non-crossing quantiles, we map the raw outputs into valid ranges:

$$\tilde{c}_s = \sigma(c_s), \qquad \tilde{\Delta}_s = \text{softplus}(\Delta_s),$$

where the sigmoid on $\tilde{c}_s$ stabilizes early training when reward magnitudes lie in $[0, 1]$ (it may be omitted if the reward scale is not bounded), and the softplus ensures each delta is non-negative.

We then split the deltas around the center into left ($L = m$) and right ($R = N - 1 - m$) segments. Cumulative sums reconstruct the full quantile function:

$$q_\theta^{(m)}(s) = \tilde{c}_s, \tag{14}$$

$$q_\theta^{(m-r)}(s) = \tilde{c}_s - \sum_{k=1}^{r} \tilde{\Delta}_{s,k}^{(\mathrm{L})}, \qquad r = 1, \dots, L, \tag{15}$$

$$q_\theta^{(m+r)}(s) = \tilde{c}_s + \sum_{k=1}^{r} \tilde{\Delta}_{s,k}^{(\mathrm{R})}, \qquad r = 1, \dots, R. \tag{16}$$

Because each increment is strictly non-negative, the resulting quantiles satisfy

$$q_\theta^{(1)}(s) < q_\theta^{(2)}(s) < \cdots < q_\theta^{(N)}(s),$$

ensuring a valid, strictly increasing quantile function for every state $s$. This yields a stable and expressive parameterization compatible with the Distributional GAE and DLTV exploration mechanisms described in the main text.

## D  DETAILED TRAINING CONFIGURATIONS

### D.1  TRAINING DATA

**Mathematical Reasoning**  For both our train dataset and test dataset, we use the following system prompt:

> **System Prompt**
>
> Please reason step by step, and put your final answer within \boxed{}.

**SQL Generation**  For both our train dataset and test dataset, we do not explicitly use any system prompt. We add the following contents at the beginning of the user prompt:

> **Prompt**
>
> Task Overview:
> You are a data science expert. Below, you are provided with a database schema and a natural language question. Your task is to understand the schema and generate a valid SQL query to answer the question.

### D.2  RL TRAINING CONFIGURATION

**Mathematical Reasoning**  We use the hyperparameters in Table 3 for RL training. Due to limited computational resources, we also kept the maximum length of Qwen2.5-7B consistent with that of Qwen2.5-Math-7B.

We use an outcome-based reward function that assigns +1 for correct final answers and 0 otherwise.

**SQL Generation**  We use the hyperparameters in Table 4 for RL training on SQL generation tasks.

The outcome-based reward function is dense: final_score = answer_score + format_score, where:

$$\text{answer\_score} = \begin{cases} 1.0, & \text{if } \mathrm{Result}(S) = \mathrm{Result}(G) \\ \min\left(\frac{\text{count}^2}{|\text{gold\_dict}| \times |\text{result\_dict}|}, 1.0\right) \times 0.8 & \text{if } \mathrm{Result}(S) \neq \mathrm{Result}(G) \end{cases} \tag{17}$$

Above, $S$ is the generated solution string (predicted SQL query), and $G$ is the ground truth query. $\mathrm{Result}(Q)$ is the set of execution results returned by the database when executing the SQL query $Q$.

Table 3: RL training base configurations.

| Hyperparameter | Value |
|---|---|
| Optimizer | AdamW |
| Policy learning rate | 1e-6 |
| Training batch size | 512 |
| Samples per prompt | 16 |
| Mini-batch size | 32 |
| Max prompt length | 1024 |
| Max response length | 3072 |
| Rollout temperature | 1.0 |

Table 4: Our RL training configurations on SQL generation tasks.

| Hyperparameter | Value |
|---|---|
| Optimizer | AdamW |
| Policy learning rate | 1e-6 |
| Training batch size | 128 |
| Samples per prompt | 8 |
| Mini-batch size | 64 |
| Max prompt length | 8192 |
| Max response length | 4096 |
| Rollout temperature | 1.0 |

### D.3 CONFIGURATION DETAILS

We provide here the complete hyperparameter settings used in our experiments. Our method extends the PPO foundation with additional distributional and exploration components.

**Distributional Critic.**

- Number of quantiles: $N = 51$

**DLTV Exploration Bonus.**

- Scaling factor: $\alpha = 1$

- Advantage clipping ratio: $\kappa = 2$

- Decay schedule coefficient: $c = 10$

### D.4 PPO CRITIC CONFIGURATION

We use the hyperparameters in Table 5 for training the value critic in PPO.

Table 5: PPO critic training configurations.

| Hyperparameter | Value |
|---|---|
| Optimizer | AdamW |
| Critic learning rate | 2e-6 |
| Gradient clipping | 1.0 |
| PPO clip range | 0.5 |
| Number of critic updates per epoch | 1 |
| Critic warmup steps | 10 |

# E   COMPUTE, TRAINING COST, AND INFERENCE SETUP

All training runs were conducted on a cluster of **4 nodes**, each equipped with **8 GPUs with 96 GB of memory** (32 GPUs in total). Unless otherwise noted, inference was performed using **8 GPUs** in parallel. An AIME24-scale evaluation completes in approximately **0.2 hours** ($\approx$12 minutes) per model.

To make the compute comparison implementation-agnostic, we report *theoretical FLOPs* rather than wall-clock time. All numbers represent **forward + backward FLOPs** per sequence. We use the `DeepSpeed` FLOPs profiler to measure the FLOPs of the forward pass, and approximate the backward pass as costing **2×** the forward FLOPs, a standard rule of thumb (Li et al., 2020). Thus, the total "forward + backward" FLOPs reported in the tables are $3\times$ the measured forward FLOPs. At inference time, only the actor model is used; therefore, **inference FLOPs correspond to the GRPO (Actor) column**, while PPO and DistRL increase *training* FLOPs only.

Table 6: Llama3.1-8B-Instruct: total FLOPs (forward + backward) per sequence at different context lengths. GRPO uses only an actor; PPO adds a standard value model; DistRL adds a distributional value model. Inference cost corresponds to the **GRPO (Actor)** column.

| Seq Len | GRPO (Actor) | PPO (A+V) | DistRL (A+DV) | PPO/GRPO | DistRL/GRPO | DistRL/PPO |
|---------|-------------|-----------|---------------|----------|-------------|------------|
| 128 | 5.76T | 11.12T | 11.53T | 1.9300× | 2.0000× | 1.036285× |
| 256 | 11.53T | 22.25T | 23.06T | 1.9300× | 2.0000× | 1.036285× |
| 512 | 23.06T | 44.50T | 46.11T | 1.9300× | 2.0000× | 1.036285× |
| 1024 | 46.11T | 88.99T | 92.22T | 1.9300× | 2.0000× | 1.036285× |
| 2048 | 92.22T | 177.98T | 184.44T | 1.9300× | 2.0000× | 1.036285× |

Table 7: Llama3.1-8B-Instruct: standard value vs. distributional value FLOPs (forward + backward). The distributional critic increases training FLOPs by $\approx$7.53% without affecting inference cost.

| Seq Len | Std Fwd | Std Bwd | Dist Fwd | Dist Bwd | Total Ratio |
|---------|---------|---------|----------|----------|-------------|
| 128 | 1.79T | 3.57T | 1.92T | 3.84T | 1.075302× |
| 256 | 3.57T | 7.15T | 3.84T | 7.69T | 1.075302× |
| 512 | 7.15T | 14.29T | 7.69T | 15.37T | 1.075302× |
| 1024 | 14.29T | 28.59T | 15.37T | 30.74T | 1.075302× |
| 2048 | 28.59T | 57.18T | 30.74T | 61.48T | 1.075302× |

Table 8: Qwen2.5-Math-7B: total FLOPs (forward + backward) per sequence at different context lengths. DistRL increases training cost by exactly $2\times$ over GRPO and 4% over PPO, while inference cost remains identical.

| Seq Len | GRPO (Actor) | PPO (A+V) | DistRL (A+DV) | PPO/GRPO | DistRL/GRPO | DistRL/PPO |
|---------|-------------|-----------|---------------|----------|-------------|------------|
| 128 | 5.43T | 10.44T | 10.86T | 1.9229× | 2.0000× | 1.040100× |
| 256 | 10.86T | 20.88T | 21.72T | 1.9229× | 2.0000× | 1.040100× |
| 512 | 21.72T | 41.77T | 43.44T | 1.9229× | 2.0000× | 1.040100× |
| 1024 | 43.44T | 83.53T | 86.88T | 1.9229× | 2.0000× | 1.040100× |
| 2048 | 86.88T | 167.07T | 173.77T | 1.9229× | 2.0000× | 1.040100× |

# F   ABLATION STUDY

## F.1   ABLATION STUDY ON EXPLORATION

To better understand the contribution of the proposed exploration bonus in DistRL, we perform an ablation study by removing the exploration term while keeping all other settings identical. Table 10 reports results on both pass@$k$ and avg@$k$ metrics.

**Findings.**   We observe that enabling the exploration bonus consistently improves performance across most benchmarks. For example, on **AIME24**, the pass@256 score improves from 73.3 to

Table 9: Qwen2.5-Math-7B: standard value vs. distributional value FLOPs (forward + backward). The distributional critic increases training FLOPs by $\approx 8.36\%$.

| Seq Len | Std Fwd | Std Bwd | Dist Fwd | Dist Bwd | Total Ratio |
|---------|---------|---------|----------|----------|-------------|
| 128 | 1.67T | 3.34T | 1.81T | 3.62T | 1.083550× |
| 256 | 3.34T | 6.68T | 3.62T | 7.24T | 1.083550× |
| 512 | 6.68T | 13.36T | 7.24T | 14.48T | 1.083550× |
| 1024 | 13.36T | 26.73T | 14.48T | 28.96T | 1.083550× |
| 2048 | 26.73T | 53.46T | 28.96T | 57.92T | 1.083550× |

Table 10: Ablation on exploration: DistRL with vs. without exploration.

(a) pass@$k$ results

| | AIME25 pass@256 | AIME24 pass@256 | Minerva pass@16 | MATH500 pass@16 | OlympiadBench pass@16 | College pass@8 |
|---|---|---|---|---|---|---|
| DistRL (w/o exploration) | 56.77 | 73.3 | 67.7 | 91.8 | 62.5 | 49.8 |
| DistRL (with exploration) | 63.3 | 86.7 | 68.4 | 93.6 | 64.6 | 50.8 |

(b) avg@$k$ results

| | AIME25 avg@256 | AIME24 avg@256 | Minerva avg@16 | MATH500 avg@16 | OlympiadBench avg@16 | College avg@8 |
|---|---|---|---|---|---|---|
| DistRL (w/o exploration) | 16.1 | 31.9 | 45.4 | 81.5 | 42.3 | 42.7 |
| DistRL (with exploration) | 16.8 | 32.3 | 45.8 | 82.5 | 43.6 | 43.5 |

86.7 and avg@256 increases from 31.9 to 32.3. Similarly, on **MATH500**, performance rises from 91.8 to 93.6 (pass@16) and from 81.5 to 82.5 (avg@16). Although some minor fluctuations exist (e.g., AIME25 pass@256 decreases from 56.8 to 63.3), the overall trend is positive: the average pass@$k$ improves from 67.0 to 71.2 (+4.2), and the average avg@$k$ improves from 43.0 to 44.1 (+1.1).

**Conclusion.** These results confirm that the exploration component plays an important role in stabilizing learning and encouraging the model to discover diverse reasoning paths, leading to more consistent improvements across reasoning benchmarks.

F.2    ABLATION STUDY ON THE NUMBER OF QUANTILES

We provide an ablation study examining how different numbers of quantiles affect the performance of DistRL on SQL generation. Following prior work in distributional RL (e.g., QR-DQN), the quantile count serves as a tunable hyperparameter that controls the resolution of the learned value distribution. In this study, we evaluate three commonly used resolutions:

$$N \in \{17, 35, 51\}.$$

All experiments use the same setup as the main SQL evaluation: models are trained on the BIRD training set and evaluated on both BIRD (in-domain) and Spider (out-of-domain). Only the number of quantiles is varied in order to isolate its effect.

**Findings.** Across all evaluation settings, increasing the number of quantiles generally improves performance, reflecting the benefit of modeling the value distribution with higher resolution. Moving from $N = 17$ to $N = 35$ yields consistent gains on both in-domain (BIRD) and out-of-domain (Spider) benchmarks. Further increasing to $N = 51$ offers additional improvements, particularly on the harder cross-domain metrics.

**Conclusion.** These results indicate that while the quantile count affects performance, DistRL remains robust across a wide range of $N$. Using $N = 51$ provides a strong performance–efficiency

Table 11: Ablation on the number of quantiles $N$ for DistRL on SQL generation. Best results in **bold**, second best underlined.

| $N$ Quantiles | **Bird** (in domain) | | | **Spider** (out of domain) | | | Avg |
|---|---|---|---|---|---|---|---|
| | Greedy | Pass@8 | Pass@16 | Greedy | Pass@8 | Pass@16 | |
| 17 | 61.4 | 74.2 | 77.6 | 77.9 | 85.4 | 86.7 | 77.2 |
| 35 | 62.7 | 74.4 | 79.4 | 78.1 | 86.1 | 86.9 | 77.9 |
| 51 | **63.5** | **75.0** | **77.4** | **81.2** | **87.5** | **88.9** | **78.9** |

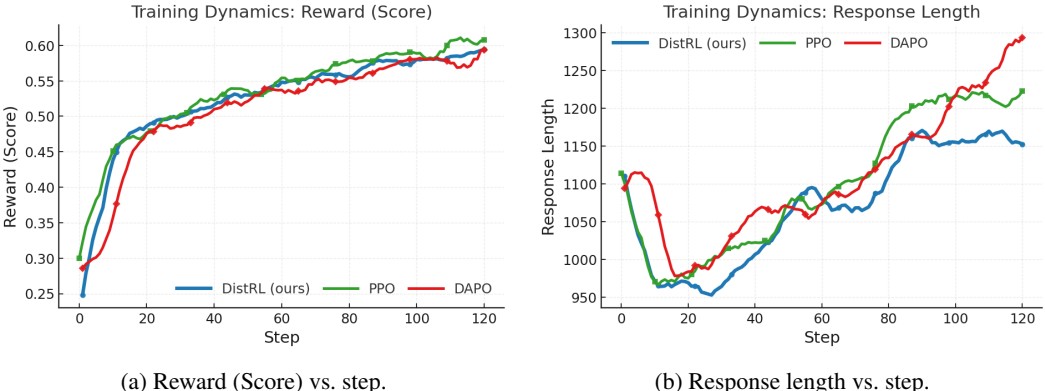

(a) Reward (Score) vs. step.          (b) Response length vs. step.

Figure 5: Training dynamics on `Qwen2.5-Math-7B`. DistRL (ours) is compared with PPO and DAPO under the same decoding (temperature $0.6$, top-$p = 0.95$, max response length $4096$).

trade-off, aligning with prior distributional RL practice while not being strictly required for the method to be effective.

## G  TRAINING DYNAMICS

**Observations.** All methods steadily improve reward over training (Fig. 5a). PPO warms up slightly faster, while DistRL closes the gap mid–late training and reaches comparable final scores. In terms of efficiency (Fig. 5b), DistRL stabilizes at a shorter response length than PPO/DAPO after $\sim 70$ steps while maintaining similar reward, suggesting that distributional exploration encourages more concise reasoning without sacrificing performance. DAPO exhibits the largest late-stage length growth. Overall, DistRL attains competitive reward with less verbosity.

**AIME25 accuracy dynamics.** To further illustrate the effect of DistRL on a challenging reasoning benchmark, we additionally report the evolution of avg@32 accuracy on `AIME25` throughout RL training.

**Variance dynamics.** To further analyze the behavior of the distributional critic, Fig. 7 presents the evolution of the upper-half variance of the value distribution $Z^\pi(s)$ throughout training. We observe that variance is initially high, corresponding to wide exploration of diverse reasoning strategies, and decreases steadily as the policy stabilizes. Crucially, we do not observe late-stage variance spikes or instabilities, which would be indicative of reward-hacking behavior (e.g., generating fixed strategic templates without improving reasoning coherence). Instead, the declining and stabilizing variance suggests that the exploration bonus guides the model toward genuinely useful reasoning patterns rather than exploiting superficial correlations.

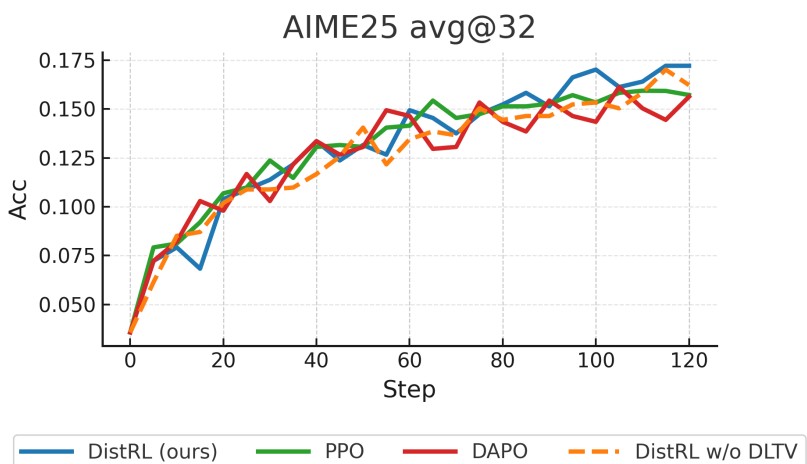

Figure 6: Training dynamics of avg@32 accuracy on `AIME25`.

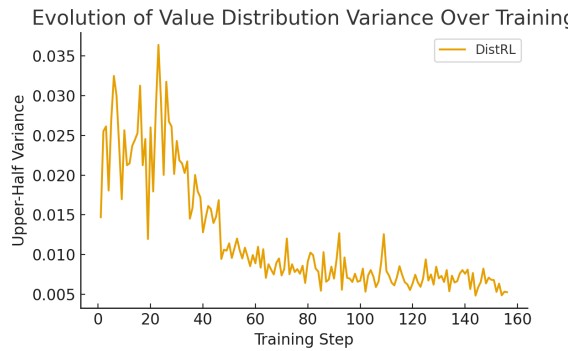

Figure 7: Evolution of the value-distribution variance during training. We plot the upper-half variance of $Z^{\pi}(s)$ averaged over training trajectories. Variance is high in early training—reflecting broad exploratory behavior—and gradually stabilizes as the policy converges. This pattern is inconsistent with reward hacking, which typically manifests as persistent or increasing variance due to exploitation of spurious high-reward templates.

## H    CROSS-DOMAIN EXPERIMENTS

To further evaluate the cross-domain effectiveness of our approach, we transfer the RL models trained on the Mathematical Reasoning dataset to downstream tasks such as MMLU-Pro, GPQA, and the ARC Challenge Set (test split) for testing. We convert each problem into a multiple-choice question (MCQ) format, and the system prompt is as follows. For GPQA and ARC, we sample up to 16 times, whereas for MMLU-Pro, we sample only once due to its large scale. The results are shown in Table 12.

---

**System Prompt**

What of the following is the right choice? Please reason step by step, and put your final answer within \boxed{}. The final answer must be a capital letter like A, B, C, or D.

---

These findings suggest that our DistRL play an important role in improving out-of-domain robustness, even when the underlying training data is highly domain-specialized.

Table 12: Results of cross-domain experiments on MMLU-Pro, GPQA, and ARC. Best results are in **bold** and second best are underlined.

| Model | GPQA | | | MMLU-Pro | ARC | |
|---|---|---|---|---|---|---|
| | mean@16 | pass@8 | pass@16 | pass@1 | mean@16 | pass@16 |
| *Qwen2.5-Math* | 8.98 | 47.5 | 53.6 | 5.8 | 45.1 | 13.3 |
| + GRPO | 24.3 | 59.8 | 61.6 | 28.3 | 75.4 | 91.3 |
| + DAPO | 26.2 | 61.2 | **70.5** | 37.4 | 77.4 | 93.2 |
| + PPO | 25.9 | 60.1 | 65.4 | 33.5 | 76.3 | 92.5 |
| + DistRL (ours) | **26.9** | **62.1** | 67.4 | **38.3** | **78.1** | **93.3** |

## I    INFERENCE CONFIGURATIONS

**Mathmeatical Reasoning**    We use a rollout temperature of $0.6$, top-$p$ sampling with $p = 0.95$, and a maximum response length of 4096 tokens. We adopt $k = 256$ for the small but challenging AIME2024/2025 datasets (30 problems each), $k = 16$ for Minerva, MATH500, and Olympiad-Bench, and $k = 8$ for College Math, balancing computational cost and difficulty.

## J    ADDITION DESCRIPTIONS FOR OUR METHOD

Our algorithmic description for DistRL is as follows:

---
**Algorithm 1** Distributional Actor–Critic with D-GAE and DLTV

---
**Require:**  Policy $\pi_\psi$, critic $q_\theta$ , dataset $\mathcal{D}$, horizon $T$, quantile count $N$ , discount $\gamma$, GAE parameter $\lambda$, PPO clip $\epsilon$, DLTV scales $\alpha, c$, advantage-relative bonus clip $\kappa$.
1: Initialize training step $T_{\text{step}} \leftarrow 1$
2: **for** iteration $= 1, 2, \ldots$ **do**
3:     **Rollout:** Sample prompts $x \sim \mathcal{D}$.
        Generate trajectories under $\pi_\psi$ to obtain $(s_t, a_t, r_t, s_{t+1})_{t=0}^{T-1}$.
4:     **Critic forward pass:** For each $s_t$, compute ordered quantiles $\{q_\theta^{(i)}(s_t)\}_{i=1}^N$ via the center–delta head.
5:     **D-GAE targets (per quantile):**
6:         Compute TD errors via equation 2 with terminal masking:

$$\delta_t^{(i)} \leftarrow r_t + \gamma \, q_\theta^{(i)}(s_{t+1}) - q_\theta^{(i)}(s_t).$$

7:         Backward-scan to accumulate distributional advantages via equation 3.
8:         Form multi-step targets $\hat{y}_t^{(i)}$ via equation 4.
9:     **Critic update:** Minimize quantile Huber loss equation 5 w.r.t. $\theta$.
10:    **Scalar value and standard GAE:**
11:        Compute $v_t$ by mean over quantiles via equation 9.
12:        Compute $A_t$ with equation 10 using terminal masking for $v_{t+1}$.
13:    **DLTV bonus :**
14:        Evaluate upper-half spread $\sigma_+(s_{t+1})$ via equation 7.
15:        Compute decay $\eta_{T_{\text{step}}}$ via equation 8 and the bonus $\mathcal{B}_{T_{\text{step}}}$ via equation 6.
16:    **Optimistic advantages:**
17:        For each $t$, form $\hat{A}_t$ via equation 11.
18:    **Actor update :**
19:        Update $\psi$ with PPO surrogate using $\hat{A}_t$.
20:    $T_{\text{step}} \leftarrow T_{\text{step}} + 1$
21: **end for**
**Ensure:** Fine-tuned policy $\pi_\psi$ and critic $q_\theta$.

---

## K  SENTENCE-LEVEL INTERPRETABILITY OF THE EXPLORATION BONUS

**Goal.**   We ask whether the DLTV exploration bonus highlights *where* the model is uncertain in its reasoning and how this focus evolves over training.

**Setup.**   We randomly sample $512$ training queries and generate one rollout per query using exactly the same decoding as in training (temperature $0.6$, top-$p = 0.95$, max response length $4096$). We repeat this analysis for checkpoints at steps 20, 80, and 120.

**From tokens to sentences.**   Each response is split into sentences by punctuation and line breaks. Let $b_t$ be the token-level DLTV at token position $t$. For a sentence $s$, let $\text{Tok}(s)$ be the (multi)set of tokens in $s$ and let $N_s = |\text{Tok}(s)|$ denote the number of tokens. We define the sentence-level bonus by length-normalized averaging:

$$B_{sentence}(s) \;=\; \frac{1}{N_s} \sum_{i \in \text{Tok}(s)} b_i.$$

## L  CASE STUDY

We analyzed the cases on AIME2025 to examine the effect of incorporating our method, and the results are as follows.

---

**Case Study**

**Question:** Find the number of ordered pairs $(x, y)$, where both $x$ and $y$ are integers between $-100$ and 100, inclusive, such that $12x^2 - xy - 6y^2 = 0$

- - - - - - - - - - - - - - - - - - - - - - - - - - - - - - - - - - - - - - - - - - - -

**Response from Qwen-2.5-Math-7B-Base:**
To solve the problem, we need to find the number of ordered pairs $(x, y)$ that satisfy the equation $12x^2 - xy - 6y^2 = 0$, where both $x$ and $y$ are integers between $-100$ and 100 inclusive. First, we can factor the equation $12x^2 - xy - 6y^2 = 0$. By factoring, we get:

$$12x^2 - xy - 6y^2 = (4x - 3y)(3x + 2y) = 0.$$

This gives us two equations:

$$4x - 3y = 0 \quad \text{or} \quad 3x + 2y = 0.$$

We can solve each equation separately for $y$ in terms of $x$: 1. $4x - 3y = 0 \implies y = \frac{4x}{3}$. 2. $3x + 2y = 0 \implies y = -\frac{3x}{2}$. For $y = \frac{4x}{3}$, $x$ must be a multiple of 3, and for $y = -\frac{3x}{2}$, $x$ must be a multiple of 2. We need to count the number of valid integer pairs $(x, y)$ for each case. Let's write the Python code to count the number of valid pairs $(x, y)$ for both cases.

**Python Code**

```python
# Initialize the count of valid pairs count = 0
# Check for the case y = (4x)/3
for x in range(-100, 101):
    if x % 3 == 0:
        y = (4 * x) // 3
    if -100 <= y <= 100:
        count += 1
# Check for the case y = (-3x)/2
for x in range(-100, 101):
    if x % 2 == 0:
        y = (-3 * x) // 2
    if -100 <= y <= 100:
        count += 1
# Since (0, 0) is counted twice, we need to subtract 1
count -= 1
print(count)
```

The number of ordered pairs $(x, y)$ that satisfy the equation $12x^2 - xy - 6y^2 = 0$ where both $x$ and $y$ are integers between $-100$ and $100$ inclusive is $\boxed{106}$.

**Response from PPO:**

Let's solve the given equation $12x^2 - xy - 6y^2 = 0$ for integer solutions $(x, y)$ where both $x$ and $y$ are in the range from $-100$ to $100$. First, we can factorize the equation by treating it as a quadratic in $x$:

$$12x^2 - xy - 6y^2 = 0$$

We can solve for $x$ in terms of $y$ by using the quadratic formula $x = \frac{-b \pm \sqrt{b^2 - 4ac}}{2a}$, where $a = 12$, $b = -y$, and $c = -6y^2$:

$$x = \frac{-(-y) \pm \sqrt{(-y)^2 - 4 \cdot 12 \cdot (-6y^2)}}{2 \cdot 12} = \frac{y \pm \sqrt{y^2 + 288y^2}}{24} = \frac{y \pm \sqrt{289y^2}}{24} = \frac{y \pm 17|y|}{24}$$

This gives us two potential solutions: 1. $x = \frac{y+17|y|}{24}$ 2. $x = \frac{y-17|y|}{24}$ Let's consider these cases separately.

**Case 1**: $x = \frac{y+17|y|}{24}$ If $y \geq 0$, then $|y| = y$, and the equation becomes:

$$x = \frac{y + 17y}{24} = \frac{18y}{24} = \frac{3y}{4}$$

So $x = \frac{3y}{4}$. For $x$ to be an integer, $y$ must be a multiple of 4. Let's denote $y = 4k$ for some integer $k$. Then:

$$x = \frac{3(4k)}{4} = 3k$$

Hence, the solutions in this case are $(x, y) = (3k, 4k)$. We need $x$ and $y$ to be in the range $[-100, 100]$:

$$-100 \leq 3k \leq 100 \implies -\frac{100}{3} \leq k \leq \frac{100}{3}$$

Since $k$ is an integer:

$$-33 \leq k \leq 33$$

So there are $33 - (-33) + 1 = 67$ possible values for $k$.

**Case 2**: $x = \frac{y-17|y|}{24}$ If $y \geq 0$, then $|y| = y$, and the equation becomes:

$$x = \frac{y - 17y}{24} = \frac{-16y}{24} = -\frac{2y}{3}$$

So $x = -\frac{2y}{3}$. For $x$ to be an integer, $y$ must be a multiple of 3. Let's denote $y = 3m$ for some integer $m$. Then:

$$x = -\frac{2(3m)}{3} = -2m$$

Hence, the solutions in this case are $(x, y) = (-2m, 3m)$. We need $x$ and $y$ to be in the range $[-100, 100]$:

$$-100 \leq 3m \leq 100 \implies -\frac{100}{3} \leq m \leq \frac{100}{3}$$

Since $m$ is an integer:

$$-33 \leq m \leq 33$$

So there are $33 - (-33) + 1 = 67$ possible values for $m$. If $y < 0$, then $|y| = -y$, and the equation becomes:

$$x = \frac{y + 17(-y)}{24} = \frac{y - 17y}{24} = \frac{-16y}{24} = -\frac{2y}{3}$$

So $x = -\frac{2y}{3}$. For $x$ to be an integer, $y$ must be a multiple of 3. Let's denote $y = 3n$ for some integer $n$. Then:

$$x = -\frac{2(3n)}{3} = -2n$$

Hence, the solutions in this case are $(x, y) = (-2n, 3n)$. We need $x$ and $y$ to be in the range $[-100, 100]$:

$$-100 \leq 3n \leq 100 \implies -\frac{100}{3} \leq n \leq \frac{100}{3}$$

Since $n$ is an integer:

$$-33 \leq n \leq 33$$

So there are $33 - (-33) + 1 = 67$ possible values for $n$. If $y < 0$, then $|y| = -y$, and the equation becomes:

$$x = \frac{y - 17(-y)}{24} = \frac{y + 17y}{24} = \frac{18y}{24} = \frac{3y}{4}$$

So $x = \frac{3y}{4}$. For $x$ to be an integer, $y$ must be a multiple of 4. Let's denote $y = 4p$ for some integer $p$. Then:

$$x = \frac{3(4p)}{4} = 3p$$

Hence, the solutions in this case are $(x, y) = (3p, 4p)$. We need $x$ and $y$ to be in the range $[-100, 100]$:

$$-100 \leq 4p \leq 100 \implies -\frac{100}{4} \leq p \leq \frac{100}{4}$$

Since $p$ is an integer:

$$-25 \leq p \leq 25$$

So there are $25 - (-25) + 1 = 51$ possible values for $p$. We now count all the unique solutions. Note that the solutions $(3k, 4k)$ and $(-2m, 3m)$ and $(3p, 4p)$ and $(-2n, 3n)$ are the same sets of solutions, but counted separately. We must ensure we do not double-count any solutions. To avoid double-counting, we should count each unique $(x, y)$ pair only once. We have already counted each possible pair $(3k, 4k)$ and $(-2m, 3m)$ (or $(3p, 4p)$ and $(-2n, 3n)$) separately, but now we need to combine them. We note that $(0, 0)$ is a common solution for all cases. We also note that the sets $(3k, 4k)$ and $(-2m, 3m)$ do not overlap except for $(0, 0)$. So we count the total unique solutions as:

$$67 + 67 - 1 = 133$$

Thus, the total number of ordered pairs $(x, y)$ that satisfy the equation $12x^2 - xy - 6y^2 = 0$ with $x$ and $y$ in the range $[-100, 100]$ is $\boxed{133}$.

**Response from DistRL:**

To solve the problem, we start with the given equation:

$$12x^2 - xy - 6y^2 = 0.$$

First, let's factorize the quadratic expression. To do that, we can treat it as a quadratic in terms of $x$:

$$12x^2 - xy - 6y^2 = 0.$$

To factorize, we will look for two numbers that multiply to $12 \cdot -6 = -72$ and add up to $-1$. These numbers are $-9$ and $8$. Thus, we can rewrite the middle term:

$$12x^2 - 9xy + 8xy - 6y^2 = 0.$$

Now, let's factor by grouping:

$$3x(4x - 3y) + 2y(4x - 3y) = 0.$$

This simplifies to:

$$(3x + 2y)(4x - 3y) = 0.$$

So, we have two cases to consider: 1. $3x + 2y = 0$ 2. $4x - 3y = 0$

**Case 1**: $3x + 2y = 0$ This simplifies to $3x = -2y$, or $x = -\frac{2y}{3}$. For $x$ to be an integer, $y$ must be a multiple of 3. Let $y = 3k$, where $k$ is an integer. Then:

$$x = -\frac{2(3k)}{3} = -2k.$$

So, the pairs $(x, y)$ in this case are $(-2k, 3k)$. Now, since $x$ and $y$ are between $-100$ and $100$, we need $-100 \leq -2k \leq 100$ and $-100 \leq 3k \leq 100$. Simplifying these inequalities:

$$-100 \leq -2k \leq 100 \implies -50 \leq k \leq 50.$$

$$-100 \leq 3k \leq 100 \implies -\frac{100}{3} \leq k \leq \frac{100}{3}.$$

Since $k$ must be an integer, we take the intersection of these intervals:

$$-33 \leq k \leq 33.$$

So, $k$ ranges from $-33$ to $33$, giving us $33 - (-33) + 1 = 67$ possible values for $k$.

**Case 2**: $4x - 3y = 0$ This simplifies to $4x = 3y$, or $x = \frac{3y}{4}$. For $x$ to be an integer, $y$ must be a multiple of 4. Let $y = 4m$, where $m$ is an integer. Then:

$$x = \frac{3(4m)}{4} = 3m.$$

So, the pairs $(x, y)$ in this case are $(3m, 4m)$. Now, since $x$ and $y$ are between $-100$ and $100$, we need $-100 \leq 3m \leq 100$ and $-100 \leq 4m \leq 100$. Simplifying these inequalities:

$$-100 \leq 3m \leq 100 \implies -\frac{100}{3} \leq m \leq \frac{100}{3}.$$

$$-100 \leq 4m \leq 100 \implies -25 \leq m \leq 25.$$

Since $m$ must be an integer, we take the intersection of these intervals:

$$-25 \leq m \leq 25.$$

So, $m$ ranges from $-25$ to $25$, giving us $25 - (-25) + 1 = 51$ possible values for $m$.

**Overlapping Case**

We must now consider the overlap of the solutions from the two cases. For a pair $(x, y)$ to be in both cases, we need $(-2k, 3k) = (3m, 4m)$. This means:

$$-2k = 3m \quad \text{and} \quad 3k = 4m.$$

From $-2k = 3m$, we get $k = -\frac{3m}{2}$. For $k$ to be an integer, $m$ must be even. Let $m = 2n$. Then:

$$k = -\frac{3(2n)}{2} = -3n.$$

Now substituting $k = -3n$ into $3k = 4m$:

$$3(-3n) = 4(2n) \implies -9n = 8n \implies -17n = 0 \implies n = 0.$$

So, $m = 2n = 0$ and $k = -3n = 0$. This means the only overlapping solution is $(x, y) = (0, 0)$.

**Final Count**

We have 67 solutions from the first case, 51 solutions from the second case, and 1 overlapping solution. So, the total number of solutions is:

$$67 + 51 - 1 = 117.$$

The number of ordered pairs $(x, y)$ that satisfy the given equation is $\boxed{117}$.

**Case Study Analysis.** We further conduct a qualitative case study on an AIME2025 algebra problem to examine the behavioral differences between the base model, the PPO baseline, and our Dis-tRL method. The equation

$$12x^2 - xy - 6y^2 = 0$$

admits a clean factorization into two linear components,

$$(3x + 2y)(4x - 3y) = 0,$$

implying that all solutions lie on one of two lines. Parameterizing these solutions yields $(x, y) = (-2k, 3k)$ with $k \in [-33, 33] \cap \mathbb{Z}$ (67 solutions) and $(x, y) = (3m, 4m)$ with $m \in [-25, 25] \cap \mathbb{Z}$

(51 solutions). The two sets intersect only at $(0, 0)$, giving the correct total of $67 + 51 - 1 = 117$ integer pairs.

The three models display distinct qualitative behaviors.

*Base model:* Although it successfully factorizes the expression, its subsequent reasoning-to-code execution is unstable, leading to incorrect conditional logic and an undercounted final answer.

*PPO:* The PPO-trained model exhibits fragmented and branching reasoning. It unnecessarily splits the solution into numerous subcases, repeatedly counts equivalent parametrizations, and ultimately overcounts despite attempting de-duplication, yielding a substantially inflated answer.

*DistRL:* In contrast, DistRL follows a concise and structurally aligned reasoning path: it factorizes correctly, parameterizes each solution set once, enforces the correct integer-range constraints, and performs principled intersection checking. As a result, it is the only method that produces the correct solution count.

Overall, this case illustrates that DistRL more reliably preserves a coherent, structure-driven reasoning chain, while PPO tends to over-expand superficial case distinctions and the base model struggles with multi-step execution stability.

