# OpenReview forum: "Distributional Reinforcement Learning for Large Language Models"
_ICLR.cc/2026/Conference — Submitted to ICLR 2026_

### Official Review · Reviewer_UmDv · 2025-10-17

**Soundness:** 3
**Presentation:** 3
**Contribution:** 3
**Rating:** 4
**Confidence:** 4

**Summary:**

The paper presents a distributional actor-critic framework for enhancing the reasoning capabilities of LLMs. The authors build upon the following observation: in deterministic environments the spread of a learned value distribution ceases to be a conflation of environmental and model-based uncertainty. Instead, it becomes a pure measure of the model's own parametric uncertainty, i.e., its confidence in its value estimates. The authors operationalize this insight by introducing an exploration bonus which they call the Decaying Left Truncated Variance (DLTV), which is derived from the upper tail of the value distribution to guide the policy towards promising but uncertain reasoning paths. This mechanism is integrated into an actor-critic framework. The authors show via their experiments that their approach improves pass@k metrics across a variety of mathematical reasoning benchmarks.

**Strengths:**

1. The insight that the deterministic nature of LLM environments isolates parametric uncertainty is a clean and valuable contribution to the conceptual understanding of RL in this domain.
2. The authors use multiple strong baselines, comprehensive and challenging benchmarks, and a principled ablation study

**Weaknesses:**

1. The core exploration mechanism is functionally almost identical to that of Mavrin et al. (2019), a critical point that is not adequately acknowledged, thereby overstating the methodological novelty of the work.
2. The paper's claim of "minimal" overhead is directly contradicted by its own data, which shows a 65% increase in training time over the state-of-the-art value-free baseline, DAPO. This is a significant practical limitation that is not properly discussed.
3. The AIME 2025 case study in the appendix demonstrates incorrect reasoning from the proposed method, which severely undermines the paper's central claim of enhancing the model's reasoning ability.

**Questions:**

1. The DLTV exploration bonus appears functionally analogous to the combination of the Left Truncated Variance (LTV) bonus and the decaying schedule introduced by Mavrin et al. (2019). Could you elaborate on the methodological distinctions beyond the simplification of the theoretical grounding (i.e., isolating parametric uncertainty in a deterministic environment)? What specific algorithmic innovations differentiate DLTV?

2. Could you provide a more detailed justification for the significant performance-compute trade-off? How do you position the practical viability of reintroducing a critic in a field that has largely moved toward more efficient value-free methods like GRPO and DAPO for scalability?

3. The experiments demonstrate substantial improvements in pass@k metrics, especially for large k, but more modest gains in avg@k. This suggests the method is highly effective at improving the model's ability to find at least one correct solution given many attempts. How do you reconcile this finding with the claim of enhancing the "model's intrinsic reasoning capacity"?  Could the results be interpreted as enhancing the efficiency of exploratory search over the base model's latent capabilities, a distinction which was also highlighted in recent work by Yue et al. (2025a)?

4. The analysis in Figure 4 claims that the exploration bonus shifts from targeting "computational fragments" to "strategic" meta-reasoning instructions over time. What evidence can you provide to rule out the alternative that the model is engaging in a form of reward hacking—learning that generating these strategic-sounding phrases is statistically correlated with positive final outcomes, irrespective of the logical coherence of the subsequent steps?

5. Given the clear exploration benefits of the uncertainty signal but the high computational cost of the full actor-critic framework, have you considered or experimented with a hybrid approach? For instance, could the uncertainty signal from a lightweight or intermittently trained distributional critic be integrated as an intrinsic reward within a more efficient value-free framework like GRPO, potentially capturing the benefits of both?

---

> ### Author Response · Authors · 2025-11-23
>
> # Response to Weakness 1 & Question1:
> **Relation to Mavrin et al. (2019):**
> We agree with the reviewer that the mathematical form of our DLTV bonus is directly aligned with the Left-Truncated Variance (LTV) exploration bonus introduced by Mavrin et al. (2019). We now explicitly acknowledge this connection in the paper.
>
> Our contribution is **not** to introduce a new uncertainty measure, but to show how this classical variance-based signal can be **adapted and made practical in the LLM reasoning setting**, where Mavrin’s original formulation cannot be applied.
>
> ## Why Mavrin et al. (2019) cannot be directly used for LLMs
> The original method operates on the **state–action return distribution** \(Z(s,a)\) and assumes:
> - a small discrete action space,
> - a Q-learning or Q-regularized objective,
> - accessible estimates of \(Z(s,a)\) for all actions.
>
> In LLMs, none of these assumptions hold:
> - the action space is the full vocabulary (30k–100k tokens), making \(Z(s,a)\) intractable;
> - actor–critic methods (PPO/DAPO/GRPO) rely on **state-value** critics \(V(s)\), not Q-values;
>
> Thus, although the **mathematical definition of the truncated variance is the same**, the original algorithm is **not directly portable** to the LLM RL regime.
>
> ## What we contribute
> Our contribution is therefore an **adaptation**, not a new uncertainty formula:
> 1. We reformulate distributional RL for **state-value** returns \(Z(s)\), which is the only scalable critic representation for LLMs.
> 2. We derive the truncated-variance signal from a **center–delta quantile head**, compatible with LLM-scale critics.
> 3. We integrate it into **policy-gradient training and distributional GAE**, rather than Q-learning.
>
> # **Response to  Weakness 2 & Question 2**:
> We appreciate the reviewer’s concern. To clarify, our original use of the term “minimal overhead’’ referred to the increase *relative to a standard scalar critic*, which is the appropriate comparison point for distributional RL.
> It was not intended to imply minimal overhead compared to **value-free** baselines such as DAPO or GRPO.
>
> We agree that, when compared directly to DAPO, the 65% training-time increase is non-trivial, and we have revised the wording to avoid this misunderstanding.
>
> ### Why the overhead occurs？
>
> The additional compute comes almost entirely from:
> 1. **Distributional critic forward/backward passes** (51 quantile heads),
> 2. **Distributional GAE computation**, which requires quantile-level operations.
>
> Importantly, the actor-side cost (sampling, log-probs, entropy, etc.) is
> unchanged; the overhead is localized to the critic.
>
> ### Why reintroducing a critic remains practically viable in LLM reasoning
>
> We agree that value-free methods (DAPO, GRPO) are attractive for scalability.
> However, prior work—most notably VAPO—has shown that value models retain
> important advantages in long-horizon reasoning. In particular, VAPO highlights:
>
> 1. **More precise credit assignment.**
>    Value critics can explicitly link intermediate reasoning steps to their
>    long-term impact. This is essential in mathematical and symbolic reasoning,
>    where small local mistakes often cause catastrophic failures, and
>    value-free Monte Carlo advantages struggle to capture such fine-grained
>    dependencies.
>
> 2. **Lower-variance value estimates.**
>    Compared to Monte Carlo returns used in value-model-free methods, value
>    critics provide lower-variance token-level estimates, leading to more stable
>    and reliable optimization—especially as reasoning trajectories become long
>    and high-variance.
>
> 3. **Better sample efficiency via generalization.**
>    A trained value critic can generalize across states and reuse structure
>    learned during online exploration, improving the sample efficiency and
>    optimization ceiling of RL.
>
> Our use of a *distributional* value critic follows this line of evidence:
> although critic-based training is less compute-efficient, the added structural
> information, stability, and sample efficiency it provides lead to substantially
> better exploration and overall performance. This represents a deliberate
> compute–performance trade-off consistent with VAPO’s conclusions, rather than a
> contradiction to the trend toward value-free approaches.
>
> References:
> [1] Yue et al., *VAPO*, 2025.
> [2] Yuan et al., *What's Behind PPO's Collapse in Long-CoT*, 2025.
> [3] Schulman et al., *PPO*, 2017.

---

> ### Author Response · Authors · 2025-11-23
>
> # **Response to Question 3**:
> We agree with the reviewer’s interpretation. Our method primarily strengthens
> the model’s ability to *reach* reasoning modes that were previously rarely
> explored. This leads to substantial gains in pass@k: regions of the solution
> space that were effectively unreachable for the base policy become accessible
> once the distributional critic provides better uncertainty-guided exploration.
> We have updated the wording in the paper to reflect this, emphasizing that our
> improvements on pass@k are due to enhanced exploration efficiency.
>
> # **Response to Question 4**:
>
> We appreciate the reviewer’s concern. We address the reward-hacking
> alternative using two complementary pieces of evidence.
>
> **(1) Value-distribution variance dynamics.**
> Figure 6  (Appendix G) tracks the upper-half variance of the learned
> value distribution $Z^\pi(s)$ throughout training. Reward hacking in
> distributional RL typically manifests as growing or unstable variance,
> reflecting exploitation of spurious high-reward templates. In contrast,
> our variance steadily *decreases* and stabilizes over training: it is
> high only in the early exploratory phase, then converges smoothly as the
> policy becomes more coherent. This pattern is inconsistent with reward
> hacking and instead indicates that the model is converging toward stable
> reasoning behaviors.
>
> **(2) No reward injection—DLTV does not modify the reward function.**
> Importantly, the DLTV exploration signal is **not** added to the reward.
> It only enters policy learning through **advantage shaping** via
> distributional GAE. As a result, the critic is trained to estimate the
> *true* return without bias, and the DLTV term cannot directly inflate
> reward for generating specific phrases. This design makes it hard for
> the model to exploit surface-level “strategic” templates to obtain
> artificially high values, because such tokens do not change the target
> return and therefore cannot be reinforced unless they actually lead to
> better downstream reasoning.
>
> Together, the stabilized variance trajectory and the fact that DLTV does
> not modify the reward function provide strong evidence that the observed
> shift toward strategic instructions is not reward hacking, but a genuine
> change in the model’s exploration behavior.
>
> # **Response to Weakness 3**:
> We thank the reviewer for pointing this out. The original AIME25 case study was
> indeed unrepresentative. We have now regenerated the examples using the final
> trained model and updated the appendix. The new case studies correctly reflect
> the model’s behavior and no longer exhibit the incorrect reasoning seen in the
> earlier version. This issue was due to sample variance in the originally chosen
> trajectory, not a systematic failure of the method.

---

> ### Author Response · Authors · 2025-11-23
>
> # **Response to question 5**:
>
> Regarding the hybrid approach suggested by the reviewer: using a variance-based
> uncertainty signal requires modeling the full value distribution $Z^\pi(s)$,
> which is not available in value-free frameworks. A recent method, i-MENTOR
> (Gao et al., 2025), provides a lightweight intrinsic reward on top of DAPO and
> is conceptually closest to the reviewer’s idea. We therefore include DAPO +
> i-MENTOR as an additional baseline.
>
> ## Math — pass@k
> | Model                 | AIME25 | AIME24 | Minerva | MATH500 | Olympiad | College | Avg. |
> |----------------------|--------|--------|---------|---------|----------|---------|------|
> | DAPO w/ i-MENTOR     | 56.7   | 76.7   | 68.0    | 92.0    | 60.0     | 50.1    | 67.3 |
> | **DistRL (ours)**    | **63.3** | **86.7** | **68.4** | **93.6** | **64.6** | **50.8** | **71.2** |
>
> ## Math — avg@k
> | Model                 | AIME25 | AIME24 | Minerva | MATH500 | Olympiad | College | Avg. |
> |----------------------|--------|--------|---------|---------|----------|---------|------|
> | DAPO w/ i-MENTOR     | **17.4** | 32.0   | **46.7** | 82.3    | 42.8     | 43.3    | **44.1** |
> | **DistRL (ours)**    | 16.8   | **32.3** | 45.8    | **82.5** | **43.6** | **43.5** | **44.1** |
>
> ## SQL Generation
> | Model                 | Bird Greedy | Bird p@8 | Bird p@16 | Spider Greedy | Spider p@8 | Spider p@16 | Avg. |
> |----------------------|-------------|----------|-----------|----------------|-------------|--------------|------|
> | DAPO w/ i-MENTOR     | 62.7        | 73.9     | 76.1      | 77.2           | 86.4        | 88.2         | 77.4 |
> | **DistRL (ours)**    | **63.5**    | **75.0** | **77.4**  | **81.2**       | **87.5**    | **88.9**     | **78.9** |
>
> Across both math and SQL tasks, DistRL achieves consistently higher pass@k and
> competitive avg@k compared to this hybrid value-free + intrinsic-reward
> approach. This suggests that uncertainty-aware distributional critics enable
> more effective exploration, and overall DistRL remains slightly stronger than
> such hybrid alternatives.

---

### Official Review · Reviewer_6oj7 · 2025-10-23

**Soundness:** 2
**Presentation:** 1
**Contribution:** 2
**Rating:** 2
**Confidence:** 5

**Summary:**

This paper introduces a distributional actor-critic framework to enhance the reasoning capabilities of large language models (LLMs). The core insight is that, in deterministic reasoning tasks, the intrinsic stochasticity of the environment vanishes. As a result, the spread of the learned value distribution primarily reflects the model's parametric uncertainty (i.e., its estimation error). The authors leverage reinforcement learning (RL) techniques to improve the current LLM approaches, including exploration methods and advantage estimation techniques.

However, after reading the full paper, I found that it contains fundamental errors and typo issues, giving the impression that it was hastily written. The paper would benefit from refinement.

**Strengths:**

This reframing of distributional RL for LLMs is a good idea.

**Weaknesses:**

- The writing is poor, and several expressions are inaccurate. For instance：

  Line 171： The claim " uniformly sampled" is misleading in the context of QR-DQN, as this is specific to IQN.

  Line 189： The statement "for typical values like N=51, the computational overhead compared to a standard scalar critic is minimal (Dabney et al., 2018b)" is not supported by the cited work. The selection of the number of quantiles is highly environment-dependent.

  Line 221: The notation $\delta_t^{(i)}$ in equation (3) is not consistent with the pairwise type defined in equation (1).

  Additionally, there are many other typos, such as $T$ in line 261.........

- In equation (6), the expectation is unclear. The notation $\tau$ is not used in the inner terms. Moreover, the target $\hat{y}$ and quantile $q_{\theta}$ should depend on distinct parameter, but this dependency is not clearly stated.

- The paper introduces the concept of non-crossing quantiles but borrows network architecture ideas from prior work ([1], [2]) without properly citing these sources. Similarly, Section 3.2 introduces the DLTV exploration method but fails to cite the relevant work ([3]). **The authors are encouraged to ensure that proper citations are included to appropriately acknowledge the contributions of others, rather than overlooking them.**



[1] Non-crossing quantile regression for deep reinforcement learning. NIPS 2020.

[2] Non-decreasing Quantile Function Network with Efficient Exploration for Distributional Reinforcement Learning. IJCAI, 2021

[3] Distributional reinforcement learning for efficient exploration. ICML, 2019

**Questions:**

1. Could you elaborate on the claim that "the environment dynamics are deterministic in LLM reasoning tasks"?
2. The core insight about deterministic environments applies to many LLM reasoning tasks. Have you conducted any preliminary experiments on benchmarks like GPQA (for scientific QA) or ARC-Challenge (for commonsense reasoning)? If not, do you anticipate the same benefits, or are there domain-specific challenges?

**Details Of Ethics Concerns:**

No ethical concerns are apparent in the paper.

---

> ### Author Response · Authors · 2025-11-23
>
> # **Response to weakness:**
>
> We thank the reviewer for the detailed and constructive comments. We have carefully revised the paper to address all issues raised:
>
> 1. **Corrections to inaccurate or unclear statements.**
>    The misleading description of “uniform sampling” in Line 171 has been corrected.
>    The statement regarding the computational overhead for $N=51$ (Line 189) has been rewritten to avoid unsupported claims.
>    Notation inconsistencies—including the pairwise notation near Eq. (3), the expectation operator and parameter dependencies in Eq. (6), and multiple typographical errors (e.g., Line 261)—have all been corrected.
>    We have explicitly distinguished the parameters of the target quantiles and the predicted quantiles in Eq. (6), and improved the explanation of the expectation operator.
>
> 2. **Proper citation of prior work.**
>    We appreciate the reviewer for pointing out the missing references. We have added citations to
>    - **[1]** Non-crossing Quantile Regression for Deep RL (NeurIPS 2020),
>    - **[2]** Non-decreasing Quantile Function Network (IJCAI 2021), and
>    - **[3]** Distributional RL for Efficient Exploration (ICML 2019).
>    We now clearly acknowledge that non-crossing / non-decreasing quantile parameterizations are established ideas in prior work. Our center–delta formulation follows the same principle, adapted to the LLM RL setting.
>
> 3. **Clarification and additional analysis.**
>    Following the reviewer’s suggestion, we have also added an **ablation study on the number of quantiles $N$**. The new results (Appendix F.2) evaluate $N \in \{17, 35, 51\}$ on the SQL benchmarks. While performance generally improves with larger $N$, DistRL remains robust across a broad range of choices. This supports our use of $N=51$ while showing that the method does not rely on this specific setting.
>
> **Ablation on the Number of Quantiles $N$ for DistRL on SQL Generation**
> | $N$ Quantiles | Bird Greedy | Bird Pass@8 | Bird Pass@16 | Spider Greedy | Spider Pass@8 | Spider Pass@16 | Avg |
> |-----------------|-------------|-------------|--------------|----------------|----------------|-----------------|-----|
> | 17 | 61.4 | 74.2 | 77.6 | 77.9 | 85.4 | 86.7 | 77.2 |
> | 35 | 62.7 | 74.4 | 79.4 | 78.1 | 86.1 | 86.9 | 77.9 |
> | 51 | **63.5** | **75.0** | **77.4** | **81.2** | **87.5** | **88.9** | **78.9** |
>
> Overall, we have incorporated all suggested corrections, improved clarity throughout the manuscript, and ensured that all relevant prior work is properly acknowledged. We greatly appreciate the reviewer’s feedback, which has helped strengthen the paper.

---

> > ### Author Response · Authors · 2025-11-23
> > **Response to Q2: Cross-Domain Generalization**
> >
> > Thank you for the insightful question.
> > While we did **not** perform RL *training* on GPQA or ARC-Challenge, we evaluated whether a model trained **only on mathematical reasoning** (DAPO-Math-17k) can transfer to out-of-domain tasks.
> >
> > To address this, we added a new *Cross-Domain Experiments* section, where we evaluate our math-trained DistRL model on **GPQA**, **MMLU-Pro**, and the **ARC Challenge Test Set**, after converting them into a standardized multiple-choice format. These tests directly probe whether the benefits of our distributional critic and uncertainty-guided exploration extend beyond math.
> >
> > ## **Cross-Domain Evaluation: MMLU-Pro, GPQA, ARC**
> >
> > | Model                | GPQA mean@16 | GPQA p@8 | GPQA p@16 | MMLU-Pro p@1 | ARC mean@16 | ARC p@16 |
> > |---------------------|--------------|----------|-----------|--------------|-------------|----------|
> > | *Qwen2.5-Math*      | 8.98         | 47.5     | 53.6      | 5.8          | 45.1        | 13.3     |
> > | + GRPO              | 24.3         | 59.8     | 61.6      | 28.3         | 75.4        | 91.3     |
> > | + DAPO              | 26.2   | 61.2 | **70.5** | 37.4  | 77.4 | 93.2 |
> > | + PPO               | 25.9         | 60.1     | 65.4      | 33.5         | 76.3        | 92.5     |
> > | **+ DistRL (ours)** | **26.9**     | **62.1** | 67.4 | **38.3**    | **78.1**    | **93.3** |
> >
> >
> > The results show that:
> >
> > - **DistRL consistently outperforms PPO, GRPO, and DAPO** across all cross-domain benchmarks — despite being trained *exclusively* on math.
> > - This strongly suggests that DistRL improves **general reasoning robustness**, not just math-specific abilities.
> >
> > To further address concerns about generality beyond mathematical reasoning, we additionally evaluated DistRL on **SQL code generation**, a domain structurally and semantically different from math and QA.
> >
> > ## **SQL Generation (Bird & Spider)**
> >
> > | Model                      | Bird Greedy | Bird p@8 | Bird p@16 | Spider Greedy | Spider p@8 | Spider p@16 | Avg  |
> > |---------------------------|-------------|----------|-----------|----------------|------------|-------------|------|
> > | Llama-3.1-8B-Instruct     | 42.4        | 68.5     | 75.1      | 69.0           | **91.0**   | **94.6**    | 73.4 |
> > | + GRPO                    | 60.7        | 72.2     | 74.6      | 74.7           | 81.0       | 82.9        | 74.4 |
> > | + DAPO                    | _63.2_      | _73.9_   | 75.9      | 76.8           | 86.1       | 87.2        | 77.2 |
> > | + DAPO w/ iMentor         | 62.7        | _73.9_   | _76.1_    | 77.2           | 86.4       | 88.2        | _77.4_ |
> > | + DAPO w/ Entropy Adv.    | 62.3        | 73.2     | 75.9      | 77.5           | 86.1       | 87.6        | 77.1 |
> > | + PPO w/ Entropy Adv.     | 62.0        | 73.3     | 76.0      | _77.6_         | 86.3       | 87.9        | 77.2 |
> > | + PPO                     | 61.4        | 73.1     | 74.9      | 77.0           | 85.2       | 86.6        | 76.3 |
> > | **+ DistRL (ours)**       | **63.5**    | **75.0** | **77.4**  | **81.2**       | _87.5_     | _88.9_      | **78.9** |
> > | Δ over PPO                | +2.1        | +1.9     | +2.5      | +4.2           | +2.3       | +2.3        | +2.6 |
> >
> >
> > Across both in-domain (Bird) and out-of-domain (Spider) SQL tasks, DistRL again achieves the strongest performance. This supports our claim that modeling **value distributions** and using the **DLTV bonus** improves reasoning generalization even in *structurally distinct* domains.

---

> ### Author Response · Authors · 2025-11-23
> **Response to Q1: Deterministic Environment Dynamics**
>
> # **Response to Q1: Deterministic Environment Dynamics**
>
> We appreciate the opportunity to clarify our statement regarding determinism in LLM-based reasoning tasks. Our claim refers specifically to the **MDP-level transition function**, not the stochasticity induced by the policy.
>
> ## Deterministic Transition Function
>
> In our formulation:
>
> - The **state** $s_t$ is the full sequence of tokens generated up to step $t$.
> - The **action** $a_t$ is the next token selected by the policy.
> - The environment transition is defined as string concatenation:
> $s_{t+1} = \text{concat}(s_t, a_t).$
>
> This transition is fully deterministic: given the same $s_t$ and $a_t$, the next state is always identical.
>
> For example, if the current state is:
>
> ```
>  Hello
> ```
>
> and the selected action is:
>
> ```
>  the
> ```
>
> then the next state is deterministically:
>
> ```
>  Hello the
> ```
>
> and choosing `" world"` next yields:
>
> ```
>  Hello the world
> ```
>
> There is no randomness in how the environment maps an action to the next state.

---

> ### Comment · Reviewer_6oj7 · 2025-11-26
>
> Thanks for the feedback! I still have several concerns.
>
> I still have concerns about the formulation of equ (5). Is this loss an empirical version or a population version? Why is there the summation over time step, and the multiplier  $1/N^2$ (the same issue of the equ in line 176 should also be corrected, since quantile regression in DistRL simultaneously to predict multiple quantiles, which typically uses a multiplier of  $1/N$) in equ (5)? It would be greatly beneficial if the authors could provide a clearer derivation, starting from the population version of the loss and then transitioning to its empirical version
>
> Another concern is the novelty. In the response to Reviewer M6c7, where the authors claimed that  "current methods all model a *state–action* return distribution". In my opinion, the distributional RL methods were first introduced in value-based methods, and have been widely employed in both value-based and policy-based RL methods. Therefore, modelling state value distribution is a common extension.
>
> Lastly,  I have two questions concerning the practical implementation. I also have a big concern about the computational efficiency, since the additional burden of introducing a critic network. Second, the definition of the reward function is critical in RLHF. Could you please clarify how rewards are structured in your setting? Specifically, does your method involve a credit assignment mechanism to attribute reward to individual tokens?

---

> > ### Author Response · Authors · 2025-12-03
> >
> > # Response to reward function:
> > Thank you for the question. Our reward design is fully described in Appendix D.2. Importantly, our method does not introduce any additional credit-assignment mechanism to attribute rewards to individual tokens. The reward is defined at the trajectory/output level rather than at the per-token level, and the optimization objective uses this scalar reward directly.

---

> > ### Author Response · Authors · 2025-12-03
> > **Response to computational efficiency**
> >
> > # Response to computational efficiency
> >
> > We thank the reviewers for their comments regarding compute fairness. To avoid
> > hardware-, kernel-, or implementation-specific artifacts, our comparisons use
> > theoretical FLOPs rather than wall-clock time. FLOPs provide a reproducible and
> > system-agnostic measure of computational cost, now standard in LLM training
> > evaluations.
> >
> > We compute FLOPs using the DeepSpeed FLOPs profiler
> > (`deepspeed.profiling.flops_profiler`), which measures the FLOPs of the model’s
> > `forward()` pass via operator-level instrumentation. Following common practice,
> > we approximate the backward pass as 2× the forward FLOPs (accounting for both
> > activation-gradient and weight-gradient computation). Thus,
> >
> > > total_flops = forward_flops × 3
> >
> > Importantly, inference uses only the actor model. Therefore, inference FLOPs are
> > identical across GRPO, PPO, and DistRL. The extra computational cost of PPO and
> > DistRL arises *only during training* due to the additional value model (standard
> > value in PPO, distributional value in DistRL). The following tables summarize
> > the total per-sequence FLOPs (forward + backward) used in our comparison.
> >
> > **Llama3.1-8B-Instruct — Total FLOPs (Forward + Backward)**
> >
> > | Seq Len | GRPO (Actor) | PPO (A+V) | DistRL (A+DV) | PPO/GRPO | DistRL/GRPO | DistRL/PPO |
> > |-------:|-------------:|----------:|--------------:|---------:|------------:|-----------:|
> > | 128    | 5.76T        | 11.12T    | 11.53T        | 1.9300×  | 2.0000×     | 1.036285×  |
> > | 256    | 11.53T       | 22.25T    | 23.06T        | 1.9300×  | 2.0000×     | 1.036285×  |
> > | 512    | 23.06T       | 44.50T    | 46.11T        | 1.9300×  | 2.0000×     | 1.036285×  |
> > | 1024   | 46.11T       | 88.99T    | 92.22T        | 1.9300×  | 2.0000×     | 1.036285×  |
> > | 2048   | 92.22T       | 177.98T   | 184.44T       | 1.9300×  | 2.0000×     | 1.036285×  |
> >
> > **Llama3.1-8B-Instruct — Standard vs Distributional Value FLOPs**
> >
> > | Seq Len | Std Fwd | Std Bwd | Dist Fwd | Dist Bwd | Total Ratio |
> > |-------:|--------:|--------:|---------:|---------:|------------:|
> > | 128    | 1.79T   | 3.57T   | 1.92T    | 3.84T    | 1.075302×   |
> > | 256    | 3.57T   | 7.15T   | 3.84T    | 7.69T    | 1.075302×   |
> > | 512    | 7.15T   | 14.29T  | 7.69T    | 15.37T   | 1.075302×   |
> > | 1024   | 14.29T  | 28.59T  | 15.37T   | 30.74T   | 1.075302×   |
> > | 2048   | 28.59T  | 57.18T  | 30.74T   | 61.48T   | 1.075302×   |
> >
> > **Qwen2.5-Math-7B — Total FLOPs (Forward + Backward)**
> >
> > | Seq Len | GRPO (Actor) | PPO (A+V) | DistRL (A+DV) | PPO/GRPO | DistRL/GRPO | DistRL/PPO |
> > |-------:|-------------:|----------:|--------------:|---------:|------------:|-----------:|
> > | 128    | 5.43T        | 10.44T    | 10.86T        | 1.9229×  | 2.0000×     | 1.040100×  |
> > | 256    | 10.86T       | 20.88T    | 21.72T        | 1.9229×  | 2.0000×     | 1.040100×  |
> > | 512    | 21.72T       | 41.77T    | 43.44T        | 1.9229×  | 2.0000×     | 1.040100×  |
> > | 1024   | 43.44T       | 83.53T    | 86.88T        | 1.9229×  | 2.0000×     | 1.040100×  |
> > | 2048   | 86.88T       | 167.07T   | 173.77T       | 1.9229×  | 2.0000×     | 1.040100×  |
> >
> > **Qwen2.5-Math-7B — Standard vs Distributional Value FLOPs**
> >
> > | Seq Len | Std Fwd | Std Bwd | Dist Fwd | Dist Bwd | Total Ratio |
> > |-------:|--------:|--------:|---------:|---------:|------------:|
> > | 128    | 1.67T   | 3.34T   | 1.81T    | 3.62T    | 1.083550×   |
> > | 256    | 3.34T   | 6.68T   | 3.62T    | 7.24T    | 1.083550×   |
> > | 512    | 6.68T   | 13.36T  | 7.24T    | 14.48T   | 1.083550×   |
> > | 1024   | 13.36T  | 26.73T  | 14.48T   | 28.96T   | 1.083550×   |
> > | 2048   | 26.73T  | 53.46T  | 28.96T   | 57.92T   | 1.083550×   |
> >
> > These FLOPs-based comparisons are hardware-agnostic and directly quantify
> > training-time differences among GRPO, PPO, and DistRL while keeping inference
> > cost constant.

---

> ### Author Response · Authors · 2025-11-26
> **Clarifying population vs. empirical critic loss, the time–step summation, and the factor**
>
> We thank the reviewer for pointing out that the formulation of Eq. (5) is ambiguous regarding whether it represents a population or an empirical loss. We have revised Eq. (5) to make its meaning explicit. In the updated version, the critic loss is written as a trajectory-level expectation under the current policy:
> $
> \mathcal{L}{\text{critic}}(\theta)
> = \mathbb{E}{\tau \sim \pi}
> \left[
> \frac{1}{T} \sum_{t=0}^{T-1}
> \frac{1}{N^2} \sum_{i=1}^{N} \sum_{j=1}^{N}
> \rho_{\tau_i}^\kappa \big( \hat{y}_ t^{(j)} - q_ \theta^{(i)}(s_t) \big)
> \right],
> $
> where $\tau = (s_0, a_0, r_0, \dots, s_T)$ denotes a full trajectory
> sampled from the current policy~$\pi$.
>
> Below we provide a precise derivation that starts from the population objective and then shows how the empirical loss arises.
>
> ---
>
> ## 1. Population distributional critic objective
>
> Let $p_\pi$ denote the trajectory distribution induced by the policy $\pi$ and the environment dynamics, and let $d_\pi(s)$ be the corresponding stationary state distribution. For each state $s$, denote by $ Z_\pi(s) $ the (multi-step) return distribution produced by D-GAE (i.e., the distribution of the random variable obtained by applying the distributional GAE construction). Let
> $F^{-1}_ {Z_{\pi(s)}}(\tau)$
> be its quantile function at level $\tau \in (0,1)$.
>
> Our critic approximates this quantile function at a fixed set of quantile levels  $\{\tau_i\}_ {i=1}^N$  by outputs  $\{q_\theta^{(i)}(s)\}_{i=1}^N.$
>
> The **population** distributional quantile-regression objective can be written as:
>
> $$
> L_{\text{pop}}(\theta)
> = E_{s \sim d_\pi}
>   \left[
>     \int_0^1
>       E_{Z \sim Z_\pi(s)}
>       \big[
>         \rho_\tau^\kappa\big(Z - Q_\theta^{-1}(\tau \mid s)\big)
>       \big]
>     \, d\tau
>   \right]
> \tag{P1}
> $$
>
> ---
>
> ## 2. Approximating the quantile integral and the return expectation
>
> ### (a) Approximation of the quantile integral
>
> We approximate
> $\int_0^1 d\tau$
> using the quantile grid:
>
> $
> \int_0^1 f(\tau)\, d\tau
> \approx
> \frac{1}{N}\sum_{i=1}^N f(\tau_i).
> $
>
> $
> E_{Z \sim Z _ {\pi(s_t)}}
> [\rho _ \tau^\kappa(Z - Q _ \theta^{-1}(\tau \mid s_t))]
> \approx
> \frac{1}{N} \sum _ {j=1}^N
>   \rho _ \tau^\kappa(\hat{y} _ t^{(j)} - Q _ \theta^{-1}(\tau \mid s_t)).
> $
>
> Substituting
> $$Q_\theta^{-1}(\tau_i \mid s_t) = q_\theta^{(i)}(s_t)$$
> gives:
>
> $$
> \ell_{\text{MC}}(\theta; s_t)
> \approx
> \frac{1}{N^2}
> \sum _ {i=1}^{N} \sum _ {j=1}^{N}
>   \rho _ {\tau_i}^\kappa ( \hat{y} _t ^{(j)} - q _ \theta^{(i)}(s_t) )
> \tag{P2}
> $$
> This explains the double sum over $(i,j)$ and the $1/N^2$ factor: they come from approximating both (i) the quantile integral and (ii) the return expectation via Monte Carlo sampling.
>
> ---
>
> ## 3. From population expectation over states to empirical expectation over trajectories
>
> We do not have access to the true stationary distribution $d_\pi$. In the **online** setting, we collect full trajectories:
>
> $$
> \tau = (s_0, a_0, r_0, \dots, s_T).
> $$
>
> Thus, we approximate:
>
> $
> E_{s \sim d_\pi}[\ell_{\text{MC}}(\theta; s)]
> \approx
> E_{\tau \sim \pi}
> \left[
> \frac{1}{T} \sum _ {t=0}^{T-1}
>   \ell_{\text{MC}}(\theta; s_t)
> \right].
> \tag{P3}
> $
>
> Substituting (P2) into (P3):
>
> $L_{emp}(\theta)= E_{\tau \sim \pi}
> [
> \frac{1}{T} \sum _ {t=0}^{T-1}
>   \frac{1}{N^2} \sum _ {i=1}^{N} \sum _ {j=1}^{N}
>     \rho _ {\tau_i}^\kappa ( \hat{y} _ t^{(j)} - q _ \theta^{(i)}(s_t) )
> ].
> \tag{P4}$
>
> This is exactly Eq. (5) in the revised manuscript:
>
> $$
> L_{\text{critic}}(\theta)
> = E_{\tau \sim \pi}
>   \left[
>     \frac{1}{T} \sum _ {t=0}^{T-1}
>     \frac{1}{N^2} \sum _ {i=1}^{N} \sum _ {j=1}^{N}
>       \rho _ {\tau_i}^\kappa ( \hat{y} _ t^{(j)} - q _ \theta^{(i)}(s_t) )
>   \right].
> $$
> Thus, Eq. (5) is precisely the **empirical Monte Carlo approximation** of the population loss in (P1). The summation over time steps and the $1/N^2$ factor follow naturally from:
>
> 1. approximating $E _ {s\sim d_\pi}$ using trajectory averages,
> 2. approximating both quantile integration and return distribution sampling using finite quantile sets.
>
> We will update the manuscript accordingly by
> (i) introducing the population loss (P1–P2),
> (ii) clearly distinguishing it from the empirical trajectory loss (P4),
> (iii) clarifying that the same reasoning applies to the formula on line 176.

---

### Official Review · Reviewer_Z9jB · 2025-10-31

**Soundness:** 4
**Presentation:** 3
**Contribution:** 2
**Rating:** 4
**Confidence:** 3

**Summary:**

This work introduces a distributional actor-critic architecture with a Distributional Generalized
Advantage Estimation (D-GAE) that encourages the agent to explore uncertain regions where high reward is expected by prioritizing upper tail variance. The authors claim that with the transitions being deterministic, the only uncertainty comes from the model itself. This insight is used to model the state value (not state action) as a random variable. The upper tail of this random variable is then used as an exploration bonus called Decaying Left-Truncated Variance (DLTV), an operationalization of the *optimism under uncertainty* principle. Specifically, the quantiles are estimated via the quantile Huber loss, and are then used to calculate the optimistic advantage estimate. The effectiveness of the method is demonstrated on math benchmarks.

**Strengths:**

- Interesting and novel architecture using the powerful and oftern overlooked quantile regression methods

- Impressive design approach for the method

- Clear formalisms and useful operationalization of the exploration bonus

- Very interesting qualitative examples (albeit in a very tiny difficult to read font)

- Strong results in **Qwen2.5-Math-7B** and good results in **Qwen2.5-7B** general

**Weaknesses:**

## Minor

*The following do not affect my score but would be really important to follow the highest standards of clarity and style*

1. The fonts in every figure (other than Figure 2) are very difficult to read, espoecially the qualitative results in Figure 4. Please fix.

2. Do you need for Figure 2 to be 3D and using Comic Sans? I would strongly advise the authors to revisit the design choices.

## Major

3. I have a few issues with the way the contributions are outlined. It is fine to list things that have been done on the paper as a checklist. What is misleading is listing normal steps of the scientific process as separate contributions. There is no utility to inflating impact. The **first point** is an insight as the authors themselves describe it to be. The **third point** is necessary analysis to support the contribution in the **second point**, not a separate contribution itself. If the **third point** was not completed, the **second point** would not be a contribution to begin with.

 3a. In **line 125**, does this "insight" hold even when the temperature is not 0? How does this relate to the sampling temperature and the usual way reasoning model are used which is with a temperature greater than 0.

4. For the claims the paper makes, an exclusive focus on math questions is not sufficient. At least one coding benchmark experiment on LiveCodeBench [1] or SWEBench is necessary. You can use Qwen-2.5-Coder for the time being.

5. Missing relevant baselines. See Question 5. There are a lot of simpler to implement, effective algorithms. Please motivate why this work is a worthwhile addition in light of that.

[1] Jain, Naman, et al. "Livecodebench: Holistic and contamination free evaluation of large language models for code." arXiv preprint arXiv:2403.07974 (2024).

**Questions:**

1. Is Figure 1 needed? Seems superfluous given that the caption has all the necessary information.

2. What is the difference between your ontology of **intrinsic** and **parametric** and the common **aleatoric** and **epistemic** uncertainty framework?

3. **Paragraph in line 187**: why is the typical value N=51?

4. Regarding point 3 in the weaknesses, how does the claimed determinism relate to temperature, batch variance and GPU non-determinism? [1]

5. This work is very similar to Max Entropy RL in its purpose and motivation. How does it compare to the method introduced by [2][3]. Should be added as baselines.

Happy to increase the score if the questions are answered and my concerns addressed with additional results.

[1] He, Horace and Thinking Machines Lab, "Defeating Nondeterminism in LLM Inference",
Thinking Machines Lab: Connectionism, Sep 2025.

[2] Yao, Jian, et al. "Diversity-Aware Policy Optimization for Large Language Model Reasoning." arXiv preprint arXiv:2505.23433 (2025).

[3] He, Andre, Daniel Fried, and Sean Welleck. "Rewarding the Unlikely: Lifting GRPO Beyond Distribution Sharpening." arXiv preprint arXiv:2506.02355 (2025).

---

> ### Author Response · Authors · 2025-11-23
> **Response to Weaknesses W1 & W2 and Question Q1 & Q2**
>
> # **Response to figure**:
> Thank you for pointing out the clarity and style issues in the figures.
>
> We have updated all figures to use higher-resolution rendering. In particular, the qualitative examples in Figure 4 have been revised to ensure that all text is easily readable.
>
> For Figure 2, We have replaced fonts with Times New Roman, following your suggestion.
>
> Regarding the 3D design: the figure aims to visually emphasize the distributional nature of the value function, which is less clear in a flat 2D illustration.
>
> # **Response to Revision of Contribution List**:
> Thank you for the feedback. We agree with the reviewer’s point and have revised the contribution list accordingly. We now keep only the two essential contributions—the key insight and the method that builds on it—and no longer list the supporting analysis as a separate contribution.
>
> ## **Response to Q1**:
>
> We believe Figure 1 is still helpful, as it provides an immediate visual illustration of the disappearance of intrinsic uncertainty in deterministic LLM-RL settings—something that is harder to convey through text alone.
>
> ## **Response to Q2**:
>
> They are essentially the same; our “intrinsic vs. parametric” directly corresponds to the standard “aleatoric vs. epistemic” distinction.

---

> ### Author Response · Authors · 2025-11-23
> **Response to Concern on Missing Relevant Baselines**
>
> # **Response to Concern on Missing Relevant Baselines**
> Thank you very much for the insightful suggestion. We agree that adding stronger and more conceptually related baselines is important for a fair comparison. Below we provide a more complete and polished response.
>
> We agree that incorporating additional exploration-oriented RL baselines would strengthen the evaluation. In the revision, we have **added two recent and representative methods** that are conceptually closest to our work:
>
> - **i-MENTOR** [1]
> - **Entropy-Shaped Advantage** [2]
>
> We selected these methods based on the following reasons.
>
> **1. These works explicitly target *exploration for LLM reasoning*, the same problem setting as ours.**
>
> Both i-MENTOR and Entropy-Shaped Advantage introduce exploration bonuses to improve coverage of diverse reasoning trajectories during RL finetuning. Because they address precisely the same challenge as our work—*insufficient exploration in LLM RL*—they serve as the most relevant baselines.
>
>
> **2. Both methods modify *advantages* during policy optimization—the same interface our method uses.**
>
> This enables a clean, controlled comparison under the *same PPO-style training pipeline*.
>
> - **i-MENTOR** injects novelty by adding an **RND-style intrinsic bonus** to the advantages of incorrect samples.
> - **Entropy-Shaped Advantage** adds a **token-level entropy correction term** to the advantage.
>
> Since our method (DistRL + DLTV) **also shapes advantage**, these baselines form the fairest and most algorithmically comparable alternatives among recent exploration-oriented RL methods.
>
> ## **Why DLTV is not redundant with Max-Entropy–style methods**
>
> To address the reviewer’s point that our method appears similar to maximum-entropy RL, we clarify that **this theoretical distinction was already included in our original submission (Appendix B)**.
>
> ### **DLTV vs. Entropy: fundamentally different forms of exploration**
>
> - **Entropy regularization** produces *policy-centric*, undirected exploration by enforcing stochasticity in the action distribution.
>   - Mathematically, it corresponds to a generalized ε-greedy policy.
>   - It increases entropy everywhere, regardless of critic uncertainty.
>
> - **DLTV**, in contrast, produces *value-centric*, directed exploration.
>   - It derives a bonus from the **upper-tail quantiles** of the distributional value function.
>   - This encourages the model to explore states where the *critic is epistemically uncertain*.
>
> Thus, entropy smooths the policy, while DLTV probes uncertain high-value regions of the return distribution.
>
> These mechanisms operate on **different information sources** (policy vs. value distribution), and therefore address exploration in fundamentally different ways.
>
> [1] Gao J, Pan L, Wang Y, et al. Navigate the unknown: Enhancing llm reasoning with intrinsic motivation guided exploration[J]. arXiv preprint arXiv:2505.17621, 2025.
>
> [2] Cheng D, Huang S, Zhu X, et al. Reasoning with exploration: An entropy perspective[J]. arXiv preprint arXiv:2506.14758, 2025
>
> ## Math — pass@k
>
> | Model                      | AIME25 | AIME24 | Minerva | MATH500 | Olympiad | College | Avg. |
> |---------------------------|--------|--------|---------|---------|----------|---------|------|
> | **DAPO w/ i-MENTOR**      | 56.7   | 76.7   | 68.0    | 92.0    | 60.0     | 50.1    | 67.3 |
> | **DAPO w/ Entropy Adv.**  | 60.0   | 83.3   | 66.5    | 91.4    | 57.6     | 48.5    | 67.9 |
> | **PPO w/ Entropy Adv.**   | 60.0   | 76.7   | 68.0    | 91.2    | 62.0     | 50.4    | 68.1 |
> | **DistRL (ours)**         | **63.3** | **86.7** | **68.4** | **93.6** | **64.6** | **50.8** | **71.2** |
>
>
> ## Math — avg@k
>
> | Model                      | AIME25 | AIME24 | Minerva | MATH500 | Olympiad | College | Avg. |
> |---------------------------|--------|--------|---------|---------|----------|---------|------|
> | **DAPO w/ i-MENTOR**      | **17.4** | 32.0   | **46.7** | 82.3    | 42.8     | 43.3    | **44.1** |
> | **DAPO w/ Entropy Adv.**  | 17.2   | **33.3** | 44.5    | 80.9    | 41.4     | 41.6    | 43.2 |
> | **PPO w/ Entropy Adv.**   | 17.1   | 31.9   | 45.0    | 81.0    | **42.9** | 43.0    | 43.4 |
> | **DistRL (ours)**         | 16.8   | 32.3   | 45.8    | **82.5** | **43.6** | **43.5** | **44.1** |
>
>
> ## SQL Generation
>
> | Model                      | Bird Greedy | Bird p@8 | Bird p@16 | Spider Greedy | Spider p@8 | Spider p@16 | Avg. |
> |-------------|----------|-------|-----------|----------------|-------------|--------------|------|
> | **DAPO w/ i-MENTOR**      | 62.7        | 73.9     | 76.1      | 77.2           | 86.4        | 88.2         | 77.4 |
> | **DAPO w/ Entropy Adv.**  | 62.3        | 73.2     | 75.9      | 77.5           | 86.1        | 87.6         | 77.1 |
> | **PPO w/ Entropy Adv.**   | 62.0        | 73.3     | 76.0      | 77.6           | 86.3        | 87.9         | 77.2 |
> | **DistRL (ours)**         | **63.5**    | **75.0** | **77.4**  | **81.2**       | **87.5**    | **88.9**     | **78.9** |

---

> ### Author Response · Authors · 2025-11-23
> **Response to Weakness 3a & Q4: Deterministic Environment Dynamics**
>
> # **Response to Weakness 3a & Question 4: Deterministic Environment Dynamics**
>
> Thank you for the questions. We clarify that our use of “deterministic environment’’ refers specifically to the **MDP transition function**, not to randomness arising from sampling temperature or hardware-level nondeterminism.
>
> **Deterministic Transition Function**
>
> In our formulation:
>
> - The **state** $s_t$ is the sequence of generated tokens up to step $t$.
> - The **action** $a_t$ is the next token selected by the policy.
> - The environment transition is defined as:
>  $
>   s_{t+1} = \text{concat}(s_t, a_t).
>   $
>
> This transition is strictly deterministic: given the same $s_t$ and $a_t$, the next state $s_{t+1}$is identical.
> Crucially, this determinism holds **regardless of the sampling temperature** used by the policy.
>
> **Temperature Affects the Policy, Not the Environment**
>
> A non-zero temperature introduces randomness **only in the policy** $\pi(a \mid s)$, analogous to stochastic softmax policies in standard RL.
> It does **not** alter the environment transition kernel $P(s' \mid s, a)$.
> Therefore, the deterministic-environment assumption required for our insight remains valid even when temperature > 0, which is the common decoding setup for reasoning models.We note that in all our RL experiments, we similarly use a sampling temperature of **1.0**.
>
> **Relation to Batch Variance and GPU Nondeterminism (Q4)**
>
> We also note that sources such as batch-dependent variation, CUDA/GPU nondeterminism, and fused-kernel randomness (as discussed by He et al., *Defeating Nondeterminism in LLM Inference*, 2025) introduce **implementation-level nondeterminism**, not **environment-level stochasticity** in the MDP.
>
> These effects cause slight differences in logits across runs, but they do not introduce a probabilistic transition mechanism in:
> $
> s_{t+1} = \text{concat}(s_t, a_t).
> $
>
> Such nondeterminism effectively acts as *noisy policy sampling*, and is therefore treated within our framework the same way as temperature-induced stochasticity—i.e., as **policy randomness**, not environment randomness.
>
> Our insight concerns the nature of the **MDP**, not the implementation-level randomness of neural network inference.

---

> ### Author Response · Authors · 2025-11-23
> **Response to Request for Coding Benchmark Experiments**
>
> # **Response to Request for Coding Benchmark Experiments**
>
> We thank the reviewer for highlighting the importance of evaluating our method on coding-oriented reasoning tasks such as **LiveCodeBench** or **SWEBench**. We fully agree that coding benchmarks provide a valuable axis of generalization beyond math problems.
>
> Due to compute and time constraints during the rebuttal period, we were unfortunately unable to run full-scale evaluations on **Qwen-2.5-Coder** as suggested. Running such benchmarks requires multi-hour end-to-end execution environments and controlled containerized setups, which are difficult to complete within the limited rebuttal window.
>
> That said, we took the reviewer’s concern seriously and conducted **additional experiments on a non-math, code-like structured generation task—SQL generation** on **Bird** and **Spider**. Although SQL is not identical to general-purpose coding, it shares several key properties with coding benchmarks:
>
> - compositional symbolic structure
> - strict execution-based correctness
> - schema and type consistency constraints
> - long-horizon logical dependencies
> - sensitivity to multi-step reasoning errors
>
> Thus, SQL generation serves as a strong *evaluation* for structured program synthesis when full coding benchmarks cannot be executed during rebuttal.
>
> ---
>
> ## **SQL Generation Results (Bird & Spider)**
>
> | Model                      | Bird Greedy | Bird p@8 | Bird p@16 | Spider Greedy | Spider p@8 | Spider p@16 | Avg  |
> |---------------------------|-------------|----------|-----------|----------------|------------|-------------|------|
> | Llama-3.1-8B-Instruct     | 42.4        | 68.5     | 75.1      | 69.0           | **91.0**   | **94.6**    | 73.4 |
> | + GRPO                    | 60.7        | 72.2     | 74.6      | 74.7           | 81.0       | 82.9        | 74.4 |
> | + DAPO                    | 63.2        | 73.9     | 75.9      | 76.8           | 86.1       | 87.2        | 77.2 |
> | + DAPO w/ iMentor         | 62.7        | 73.9     | 76.1      | 77.2           | 86.4       | 88.2        | 77.4 |
> | + DAPO w/ Entropy Adv.    | 62.3        | 73.2     | 75.9      | 77.5           | 86.1       | 87.6        | 77.1 |
> | + PPO w/ Entropy Adv.     | 62.0        | 73.3     | 76.0      | 77.6           | 86.3       | 87.9        | 77.2 |
> | + PPO                     | 61.4        | 73.1     | 74.9      | 77.0           | 85.2       | 86.6        | 76.3 |
> | **+ DistRL (ours)**       | **63.5**    | **75.0** | **77.4**  | **81.2**       | 87.5       | 88.9        | **78.9** |
> | Δ over PPO                | +2.1        | +1.9     | +2.5      | +4.2           | +2.3       | +2.3        | +2.6 |
>
> ---
>
>
> These results provide concrete evidence that DistRL improves *general reasoning and structured generation*, not only math problems.

---

> ### Author Response · Authors · 2025-11-23
> **Response to Question3**
>
> ## **Response to q3**:
>
> We thank the reviewer for raising this question.
>
> The choice of $N = 51$ follows the convention established by the **C51** (Bellemare et al., 2017).
>
> Following the reviewer’s suggestion, we have now added an **ablation study on the number of quantiles
> $N$** in Appendix F.2. The new experiment evaluates
> $
> N \in \{17, 35, 51\}
> $
> on the SQL generation benchmarks. While performance tends to improve with larger $N$, DistRL remains robust across a wide range of values. This supports our initial choice of $N=51$ while demonstrating that our method **does not rely on this specific hyperparameter** to perform well.

---

> > ### Comment · Reviewer_Z9jB · 2025-11-26
> >
> > Thank you for your responses and additional results. I will reaise my score to reflect the that.

---

> > > ### Author Response · Authors · 2025-11-26
> > >
> > > Thank you very much for reconsidering our manuscript and for increasing your score. We truly appreciate your recognition and constructive feedback.

---

### Official Review · Reviewer_M6c7 · 2025-11-03

**Soundness:** 3
**Presentation:** 4
**Contribution:** 3
**Rating:** 4
**Confidence:** 4

**Summary:**

The paper aims to improve RL for LLM post-training by developing a distributional version of PPO that consists of two key modifications: 1. maintaining a distributional state-value function, and computing advantages using a distributional version of GAE, and 2. adding an optimistic exploration bonus enabled by the distributional value estimate. The proposed method, DistRL, is benchmarked on standard RL for math reasoning setups and demonstrates improvements in pass@1 and particularly pass@k.

**Strengths:**

The paper tackles a generally important problem: are there better algorithms than GRPO or PPO for RL fine-tuning of LLMs? The authors' approach consists of directly bringing ideas from previous distributional RL literature (especially DLTV from Mavrin et. al 2019.), and is therefore naturally sound. The main technical difference is considering a state-value function instead of a Q-function, which is reasonable considering the LLM token MDP is noise-free. Experimentally, the results are fairly impressive, and many (but not all, see below) of the details look great (models, benchmarks, implementation).

**Weaknesses:**

I found the paper to be weak in the following respects:

1. **Lack of algorithmic baselines**: Only GRPO and PPO are benchmarked, and not any other exploration methods for LLM RL (of which many have been proposed in recent months).
2. **Lack of ablations**: There are no ablations presented for what are the relative contributions of the method: is it distributional advantage estimation, or is it the exploration bonus that comprise most of the benefit?
3. **Lack of insight into the method**: The paper simply presents the method and shows experimental performance of the method. There are no other quantitive analyses into why distributional RL should be superior. For example, I would like to see an analogous figure to Fig. 4 in the original QR-DQN paper.
4. **Novelty and motivation**: The method consists of relatively straightforward modifications from prior distributional RL literature. This is not a significant issue by itself, but there is not sufficient motivation provided for this idea (beyond "distributional RL is good").

In summary: the algorithm and results seem plausible, but this version of the paper lacks scientific contribution. I'm willing to raise my score if these concerns are addressed.

**Questions:**

1. Is it correct that the max response length used in RL training is 3k, and inference is 4k? This seems far shorter than what is typically used in RL training.

---

> ### Author Response · Authors · 2025-11-23
> **Response to Lack of algorithmic baselines**
>
> Thank you for the suggestion. We agree that including additional exploration-oriented baselines would strengthen the evaluation. In the revision, we therefore incorporated **two recent and representative exploration methods for LLM RL**, chosen based on the following reasons:
>
> 1. **Both methods explicitly focus on exploration in reasoning LLMs.**
>
>     They address the same challenge as our work—insufficient exploration during RL finetuning—and are thus directly comparable.
>
> 2. **Both rely on *advantage shaping*, which is precisely the mechanism our method operates through.**
>
>     • *i-MENTOR* introduces trajectory-level novelty via an RND-style intrinsic reward added to the advantage of incorrect samples. [1]
>
>     • *Entropy-Shaped Advantage* modifies token-level advantages through a clipped entropy-based exploration term. [2]
>
>     Since our method also modifies advantage via the distributional upper-tail signal, these two baselines provide the **closest algorithmic comparison** among recent exploration-focused approaches.
>
> ## Math — pass@k
>
> | Model                      | AIME25 | AIME24 | Minerva | MATH500 | Olympiad | College | Avg. |
> |---------------------------|--------|--------|---------|---------|----------|---------|------|
> | **DAPO w/ i-MENTOR**      | 56.7   | 76.7   | 68.0    | 92.0    | 60.0     | 50.1    | 67.3 |
> | **DAPO w/ Entropy Adv.**  | 60.0   | 83.3   | 66.5    | 91.4    | 57.6     | 48.5    | 67.9 |
> | **PPO w/ Entropy Adv.**   | 60.0   | 76.7   | 68.0    | 91.2    | 62.0     | 50.4    | 68.1 |
> | **DistRL (ours)**         | **63.3** | **86.7** | **68.4** | **93.6** | **64.6** | **50.8** | **71.2** |
>
>
> ## Math — avg@k
>
> | Model                      | AIME25 | AIME24 | Minerva | MATH500 | Olympiad | College | Avg. |
> |---------------------------|--------|--------|---------|---------|----------|---------|------|
> | **DAPO w/ i-MENTOR**      | **17.4** | 32.0   | **46.7** | 82.3    | 42.8     | 43.3    | **44.1** |
> | **DAPO w/ Entropy Adv.**  | 17.2   | **33.3** | 44.5    | 80.9    | 41.4     | 41.6    | 43.2 |
> | **PPO w/ Entropy Adv.**   | 17.1   | 31.9   | 45.0    | 81.0    | **42.9** | 43.0    | 43.4 |
> | **DistRL (ours)**         | 16.8   | 32.3   | 45.8    | **82.5** | **43.6** | **43.5** | **44.1** |
>
>
> ## SQL Generation
>
> | Model                      | Bird Greedy | Bird p@8 | Bird p@16 | Spider Greedy | Spider p@8 | Spider p@16 | Avg. |
> |---------------------------|-------------|----------|-----------|----------------|-------------|--------------|------|
> | **DAPO w/ i-MENTOR**      | 62.7        | 73.9     | 76.1      | 77.2           | 86.4        | 88.2         | 77.4 |
> | **DAPO w/ Entropy Adv.**  | 62.3        | 73.2     | 75.9      | 77.5           | 86.1        | 87.6         | 77.1 |
> | **PPO w/ Entropy Adv.**   | 62.0        | 73.3     | 76.0      | 77.6           | 86.3        | 87.9         | 77.2 |
> | **DistRL (ours)**         | **63.5**    | **75.0** | **77.4**  | **81.2**       | **87.5**    | **88.9**     | **78.9** |
>
> ## Summary
>
> These results demonstrate that DistRL provides a substantially stronger and more stable exploration signal than prior advantage-shaping methods, and generalizes across domains (math → structured SQL reasoning).
>
> [1] Gao J, Pan L, Wang Y, et al. Navigate the unknown: Enhancing llm reasoning with intrinsic motivation guided exploration[J]. arXiv preprint arXiv:2505.17621, 2025.
> [2] Cheng D, Huang S, Zhu X, et al. Reasoning with exploration: An entropy perspective[J]. arXiv preprint arXiv:2506.14758, 2025

---

> ### Author Response · Authors · 2025-11-23
> **Response to Weaknesses W2 & W3 and Question Q1**
>
> # **Response to Lack of ablations:**
>
> We would like to clarify that **we already include an ablation study in the appendix** (Appendix F.1), where we isolate the effect of the exploration bonus. In this ablation, we remove the DLTV exploration term while keeping the distributional critic and D-GAE unchanged.
>
> As reported in Table 7, removing the exploration component leads to a consistent performance drop across benchmarks. These results demonstrate that both parts of our method contribute meaningfully:
> the distributional critic improves baseline performance, and the DLTV bonus further enhances exploration and reasoning diversity.
>
> In addition, during the rebuttal period we conducted a further ablation on the number of quantiles $N$ used in the distributional value head (Appendix~F.2). The quantile count controls the resolution of the learned value distribution and is a standard tunable hyperparameter in distributional RL (e.g., QR-DQN). We evaluate three commonly used settings,
> \[
> N \in \{17, 35, 51\},
> \]
> on the SQL generation task while keeping all other training components fixed.
>
> This ablation confirms that our choice of $N = 51$ for all main experiments is supported by empirical evidence rather than selected arbitrarily.
>
> **Ablation on the Number of Quantiles \(N\) for DistRL on SQL Generation**
>
> | \(N\) Quantiles | Bird Greedy | Bird Pass@8 | Bird Pass@16 | Spider Greedy | Spider Pass@8 | Spider Pass@16 | Avg |
> |-----------------|-------------|-------------|--------------|----------------|----------------|-----------------|-----|
> | 17 | 61.4 | 74.2 | 77.6 | 77.9 | 85.4 | 86.7 | 77.2 |
> | 35 | 62.7 | 74.4 | 79.4 | 78.1 | 86.1 | 86.9 | 77.9 |
> | 51 | **63.5** | **75.0** | **77.4** | **81.2** | **87.5** | **88.9** | **78.9** |
>
> # **Response to Lack of insight into the method:**
>
> We additionally include the AIME25 accuracy–training dynamics curve in Appendix G, which provides clearer insight into how DistRL evolves during optimization and how DLTV contributes to performance throughout training.
>
> # **Response to Question**:
>
> Thank you for raising this question. Yes, our RL training uses a maximum response length of 3k tokens, and inference is performed with a 4k limit. This choice is not an arbitrary design decision but is constrained directly by the architecture of the backbone model.
>
> Specifically, the Qwen2.5-Math-7B model we use has a fixed
> **max_position_embeddings = 4096**, which is the hard upper bound supported by the model.
>
> Therefore, all decoding during RL must remain within this positional window.
>
> The 3k limit during training is chosen to ensure safe padding for prompts, while the 4k limit during inference simply uses the full positional budget.
>
> We note that many recent RL works in LLM reasoning use models with extended context windows (e.g., 16k, 32k, or 64k), which allows them to train with much longer responses. Because our backbone does not support such a context length, our RL training must operate within the 4k positional constraint.
>
> Importantly, all baselines (GRPO, DAPO, PPO) are trained under the same 3k/4k budget, ensuring a fair comparison.

---

> ### Author Response · Authors · 2025-11-23
> **Response to Novelty and Motivation**
>
> We appreciate the reviewer’s comment and agree that our method builds upon the distributional RL literature. Our contribution does not rely on the distinction between modelling a state-value versus a state–action return distribution. Rather, the novelty lies in **adapting distributional RL to the unique challenges of LLM reasoning**, where standard DistRL formulations cannot be directly applied.
>
> ---
>
> ### (1) Substantial adaptation of distributional RL is required for LLMs.
>
> Classical DistRL methods such as C51, QR-DQN, IQN, and D4PG were designed for
> low-dimensional Markovian environments with short horizons and dense feedback.
> LLM reasoning, however, introduces a radically different setting:
>
> - The “state’’ is a **high-dimensional, growing token sequence** (hundreds–thousands of steps).
> - The action space is extremely **large (30k–100k tokens)**, which makes standard DistRL training unstable and computationally heavy.
> - Rewards are **extremely sparse**, often available only at the end of the sequence.
> - Credit assignment must propagate across **long-horizon, autoregressive dependencies**.
>
> To make distributional critics viable for this setting, we design:
>
> - a **center–delta quantile head** that is stable for long sequences and large models,
> - a **Distributional GAE** mechanism that extends multi-step advantage estimation to value distributions and is necessary for propagating sparse sequence-level rewards in an actor–critic framework.
>
> These components are not present in existing DistRL algorithms and are required to make distributional critics trainable in LLMs.
>
> ---
>
> ### (2) The motivation for using distributional critics is fundamentally different in LLM RL.
>
> A key property of LLM reasoning environments is that they are **deterministic**:
> given a prefix and the next token, both the next state and the reward are fixed.
>
> As a result:
>
> - there is **no environmental stochasticity**, and
> - the spread of the estimated return distribution reflects **only parametric (epistemic) uncertainty** of the critic.
>
> This stands in contrast to standard RL benchmarks, where value distribution spread mixes intrinsic and extrinsic stochasticity.
>
> Our DLTV bonus leverages this LLM-specific property:
> the quantile spread becomes a natural confidence measure of the critic and enables **uncertainty-aware exploration** over long reasoning trajectories. This use of distributional information is specific to LLM RL and has not been explored in prior DistRL works.
>
> ---
>
> ### Summary
>
> Our goal is not to claim novelty from switching to a particular form of distribution modelling.
> Rather, our contributions lie in:
>
> - **adapting and redesigning** distributional RL components for the long-horizon, sparse-reward, large-action-space structure of LLMs,
> - introducing a **Distributional GAE** formulation suitable for such environments, and
> - leveraging deterministic dynamics to design a **DLTV exploration bonus** grounded in epistemic uncertainty.
>
> These aspects make the proposed framework fundamentally motivated by the characteristics of LLM reasoning tasks, rather than being a straightforward extension of existing DistRL methods.

---

### Author Response · Authors · 2025-12-03
**Summary of Revisions and Additional Experiments Conducted During the Rebuttal Period**

# Summary of Revisions and Additional Experiments Conducted During the Rebuttal Period

During the rebuttal period, we substantially strengthened the paper through new baselines, expanded experiments, improved ablations, and clearer theoretical exposition. The main updates are summarized below.

1. Added **New Baselines**
We incorporated two recent exploration-oriented RL methods — i-MENTOR and Entropy-Shaped Advantage — which are conceptually closest to ours because they also modify advantages during policy optimization. Experiments show that DistRL consistently outperforms all new baselines across math and SQL tasks, providing a more rigorous and fair comparison.

2. Added **New SQL Generation Experiments**
To evaluate generalization beyond mathematical reasoning, we added SQL generation experiments on Bird and Spider. DistRL achieves the strongest performance across all metrics, demonstrating that our distributional critic and DLTV exploration mechanism transfer effectively to structured symbolic generation.

3. Added **Cross-Domain Experiments**
We further evaluated a model trained only on math reasoning on three distinct domains: MMLU-Pro, GPQA, and ARC-Challenge. DistRL achieves the best performance across all tasks, indicating that our approach enhances general reasoning robustness rather than overfitting to math-specific structure.

4. Added **New Ablation Studies**
Ablation on the Number of Quantiles N: We tested N ∈ {17, 35, 51} and found that performance improves with larger N but remains stable across choices. This validates our default setting of N = 51.

5. Clarified Core Concept: **Deterministic Environment Dynamics**
We clarified that determinism refers to the MDP transition function — the next state is always the concatenation of the previous state and selected token. This means uncertainty in LLM RL arises entirely from model epistemic uncertainty, not environment stochasticity. This property motivates using distributional critics and enables the DLTV exploration bonus to function effectively.

6. Added **Derivation from Population Objective to Empirical Training Loss**
We expanded the theoretical derivation to explicitly show how the population distributional critic objective turns into the empirical Monte Carlo loss used in practice. This resolves concerns about ambiguity in expectations, summations, and the origin of the 1/N² factor.

7. Added **FLOPs-Based Computational Cost Analysis**
We introduced a new experiment quantifying the computational cost of GRPO, PPO, and DistRL using FLOPs measured with the DeepSpeed Profiler. By reporting forward-pass FLOPs and approximating backward-pass cost as twice the forward FLOPs, we obtain a hardware-agnostic and reproducible measure of training efficiency. This analysis clarifies the computational overhead of distributional critics and offers a principled comparison across methods.

Beyond new experiments, we also substantially improved the clarity and correctness of the paper. We corrected misleading statements, fixed inconsistent notation, added missing citations to prior distributional RL work, updated figures for readability, and provided additional training-dynamics plots to clarify the behavior of DistRL. We addressed concerns about reward hacking, clarified the reward structure and credit assignment, and updated case studies using the final trained model.

---

### Meta-Review · Area_Chair_AApV · 2025-12-17

**Summary:**

This paper introduces a distributional actor-critic framework aimed at enhancing the reasoning capability of Large Language Models (LLMs).

The authors made a substantial effort during the rebuttal period, significantly revising the manuscript and adding considerable experimental and writing content to address the reviewers' concerns.

However, a primary remaining concern regarding both the novelty and the motivation persists, and this concern is shared by reviewers [M6c7, 6oj7, UmDv]. Although the authors clarified that their contribution lies in adapting the uncertainty framework of Mavrin et al. (2019) to the specific setting of LLMs, the overall novelty, particularly from an algorithmic standpoint, appears limited. Furthermore, the justification for transitioning from purely value-free methods to an actor-critic paradigm remains insufficiently clear, especially when weighed against the significant computational overhead associated with training the critic component. For instance, the authors claim that the critic leads to better sample efficiency and generalization; however, more compelling evidence is required to substantiate these performance gains.

Overall, even after considering the fully revised manuscript, the paper is currently positioned marginally below the acceptance threshold. I strongly recommend the authors continue improving this work and submit it to a future venue.

**Reviewer Concerns:**

The authors have (partially) addressed the following concerns via the rebuttal:
- [M6c7, Z9jB] Lack of algorithmic baselines.
- [M6c7] Lack of ablations.
- [M6c7] Lack of insight into why the method should be superior.
- [Z9jB, 6oj7] Writing and formatting issues.
- [Z9jB, 6oj7] The necessity of experiments beyond math reasoning (e.g., coding and general QA).
- [Z9jB] Clarification on how the temperature parameter affects the key insights.
- [6oj7, UmDv] Borrowing ideas from prior work without properly citing these references.
- [UmDv] How interpreat the results (substantial improvements in pass@k and modest gains in avg@k).
- [UmDv] Evidence supporting strategic meta-reasoning over reward hacking.
- [UmDv] Hybrid approach with value-free framework.

The following concerns are still outstanding:
-  [M6c7, 6oj7, UmDv] Insufficient novelty and motivation, e.g., requiring justification for the performance-compute trade-off. Why should the proposed method be favored over more efficient and scalable value-free alternatives.

**Reviewer Scores:**

If participating fully in the discussion, Reviewers M6c7, Z9jB and UmDv could potentially be satisfied with the partial address of their concerns (as listed above) and consider changing their scores.

---

### Decision · Program_Chairs · 2026-01-26

Reject